# Development of graphitic carbon nitride quantum dots-based oxygen self-sufficient platforms for enhanced corneal crosslinking

Mei Yang[1,2,7] ✉, Tingting Chen[2,7], Xin Chen[1,7], Hongxian Pan ®[1,7], Guoli Zhao[1,7], Zhongxing Chen[1], Nan Zhao[3], Qianfang Ye[2], Ming Chen[1], Shenrong Zhang[2], Rongrong Gao[2], Keith M. Meek ®[4], Sally Hayes ®[4], Xiaowei Ma[5], Xin Li[2], Yue Wu ®[1], Yiming Zhang[6], Na Kong[6], Wei Tao ®[6], Xingtao Zhou[1] ✉ & Jinhai Huang[1] ✉

Keratoconus, a disorder characterized by corneal thinning and weakening, results in vision loss. Corneal crosslinking (CXL) can halt the progression of keratoconus. The development of accelerated corneal crosslinking (A-CXL) protocols to shorten the treatment time has been hampered by the rapid depletion of stromal oxygen when higher UVA intensities are used, resulting in a reduced cross-linking effect. It is therefore imperative to develop better methods to increase the oxygen concentration within the corneal stroma during the A-CXL process. Photocatalytic oxygen-generating nanomaterials are promising candidates to solve the hypoxia problem during A-CXL. Bio-compatible graphitic carbon nitride (g-$C_3N_4$) quantum dots (QDs)-based oxygen self-sufficient platforms including g-$C_3N_4$ QDs and riboflavin/g-$C_3N_4$ QDs composites (RF@g-$C_3N_4$ QDs) have been developed in this study. Both display excellent photocatalytic oxygen generation ability, high reactive oxygen species (ROS) yield, and excellent biosafety. More importantly, the A-CXL effect of the g-$C_3N_4$ QDs or RF@g-$C_3N_4$ QDs composite on male New Zealand white rabbits is better than that of the riboflavin 5'-phosphate sodium (RF) A-CXL protocol under the same conditions, indicating excellent strengthening of the cornea after A-CXL treatments. These lead us to suggest the potential application of g-$C_3N_4$ QDs in A-CXL for corneal ectasias and other corneal diseases.

Keratoconus is a bilateral disease of the cornea that usually begins in adolescence. The incidence of keratoconus is about 1 in 2000 (with differences between ethnic groups) and is increasing year by year[1,2]. The condition is characterized by progressive thinning of the corneal stroma and local conical protrusion of the cornea. The typical manifestations of keratoconus include myopia and irregular astigmatism resulting in blurred vision, glare and other visual problems, with the risk of blindness[1]. In very advanced cases (affecting 10–20% of keratoconus patients), corneal transplantation is required due to corneal scarring and a loss of tissue transparency[2,3]. Keratoconus remains one

of the most common reasons for corneal transplantation, while the shortage of cornea donors is problematic for both patients and society as a whole[2,3].

Corneal cross-linking (CXL) has been proven to effectively slow down or even stop the progression of keratoconus[4–6]. During CXL, riboflavin-5-phosphate (RF) is usually served as a photosensitizer. When irradiated with UVA, the photochemical reaction triggers the formation of covalent cross-links within the extracellular matrix of the corneal stroma and leads to improved corneal biomechanical strength and increased resistance to enzymatic digestion, thus

preventing further corneal thinning, deformation and vision deterioration[7,8]. Besides this, RF has the added benefit that it is a UVA absorber and can therefore limit UVA penetration into the corneal stroma and thereby prevent damage to the deeper tissues of the eye[9]. CXL therapy can avoid the need for corneal transplantation by delaying or even preventing the progress of keratoconus. As described by Wollensak et al., the conventional CXL (C-CXL) protocol involves removing the central 7–9 mm of corneal epithelium, subsequently treating the deepithelialized cornea with 0.1% RF in 20% dextran for 30 min, and exposing it to UVA irradiation (3 mW cm$^{-2}$) for another 30 min, resulting in a lengthy treatment time of approximately 1 h[10]. Based on the Bunsen-Roscoe Law of reciprocity, accelerated corneal crosslinking (A-CXL) protocols were developed which aimed to achieve the same level of cross-linking in a shorter period of time through the use of higher UVA intensities[11]. To date, Numerous studies have shown that A-CXL can stabilize, and in some cases, reduce the maximum curvature of the cornea (Kmax)[12–15]. However, the equivalence of the A-CXL and C-CXL protocols is still controversial as it is known that photosensitizers, UVA and the oxygen ($O_2$) concentration in the stroma can all affect corneal crosslinking[16]. Kamaev et al.'s study indicated that during CXL, the $O_2$ dissolved in the stroma is exhausted within the first 15 s of exposure to 3 mW cm$^{-2}$ UVA irradiation, and it takes about 10 min for the $O_2$ concentration in the stroma to start to increase again, with only 1/10 of the initial $O_2$ concentration being recovered after 30 min. In the case of A-CXL, there is a much more rapid consumption of $O_2$, with stromal depletion of $O_2$ occurring within the first 5 s of exposure to 30 mW cm$^{-2}$ UVA[17]. The importance of $O_2$ in the CXL process was further emphasized in a study conducted by Richoz et al. revealing that the biomechanical strength of pig corneas cross-linked in a hypoxic environment (using a 10 min exposure to 9 mW cm$^{-2}$ UVA) was significantly lower than that of corneas cross-linked in an atmospheric oxygen environment, underscoring that the $O_2$ concentration within the stroma is an essential limiting factor in the CXL photochemical reaction[18]. Thus, increasing the $O_2$ supply during the CXL process is regarded as a promising way of enhancing the CXL effect, driving the development of various contact and non-contact $O_2$ supply devices, as well as techniques for injecting $O_2$ into the anterior chamber[19–22]. However, laboratory studies indicate that performing A-CXL in an oxygen rich environment does not lead to a stiffer cornea, due to the slow diffusion of $O_2$ into the corneal stroma and the limited treatment time of the A-CXL procedure[23]. Therefore, there is an urgent need to devise an efficient $O_2$ supply protocol, especially one that can rapidly produce $O_2$ within the corneal stroma under the irradiation of UVA light utilized in CXL. This innovative approach could effectively address the hypoxia challenges encountered during A-CXL.

With the development of nanomaterials and nanotechnologies, nanomedicine has displayed remarkable advantages in the treatment of a number of diseases. Biocompatible photocatalytic oxygen generation nanomaterials, especially those that can produce oxygen during UVA irradiation, show great potential for overcoming the hypoxia problem during the A-CXL procedure. Being one of famous photocatalytic oxygen generation materials, graphitic carbon nitride (g-$C_3N_4$) has been widely studied and applied in photocatalytic water decomposition and solar energy conversion[24,25]. Its simple synthesis process, non-toxic nature, appropriate electron energy level structure, and stable physical and chemical properties render g-$C_3N_4$ an excellent candidate. The highest occupied molecular orbital (HOMO) and lowest unoccupied molecular orbital (LUMO) potentials of g-$C_3N_4$ are +1.4 V and −1.3 V (vs NHE, pH = 7), respectively. These potentials align with the thermodynamic requirements of photocatalytic decomposition of water to generate hydrogen and oxygen, considering that the oxidation potential for oxygen evolution in water is +0.88 V and the reduction potential for hydrogen evolution in water is −0.41 V (vs NHE, pH = 7)[26]. Many studies have indicated that g-$C_3N_4$ can be used in photocatalytic decomposition of water to produce oxygen[26,27]. In addition, the absorbance of g-$C_3N_4$ covers the range between 200 nm and 450 nm, which includes the wavelength of UVA light used for CXL (365 nm)[28]. Therefore, there is a significant interest in exploring the photocatalytic oxygen generation capacity of g-$C_3N_4$ using the same CXL UVA light. Furthermore, it is known that the electronic structure of g-$C_3N_4$ is a graphite plane formed by a homotriazine unit, which is capable of absorb UV-visible light to generate reactive oxygen species[29]. Nano-sized g-$C_3N_4$ (especially g-$C_3N_4$ quantum dots, g-$C_3N_4$ QDs) has been used in the biomedical field due to its good biocompatibility, excellent photo-luminescence, and easy surface modification properties[26,27,29–32]. Currently, g-$C_3N_4$ QDs have been reported as $^1O_2$ donors for cancer photodynamic therapy[29,32], but g-$C_3N_4$ QDs have not previously been considered as potential photosensitizers for CXL. Thus, combining their photocatalytic oxygen production capabilities and light absorbance ranges, it is of great interest to investigate the application of g-$C_3N_4$ QDs (as an independent photosensitizer solution and in conjunction with the traditionally used riboflavin solution) in the CXL process, to study its biosafety, CXL effect and possible mechanism, and establish the potential of a photosensitizer to improve the current CXL protocols.

In this work, a g-$C_3N_4$ QDs-based oxygen self-sufficient nanoplatform is developed, which comprises g-$C_3N_4$ QDs and RF@g-$C_3N_4$ QDs composite photosensitizers. Figure 1 provides a simple schematic diagram of the CXL application of g-$C_3N_4$ QDs. The photocatalytic oxygen generation capability, $^1O_2$ yield and biocompatibility of the synthesized g-$C_3N_4$ QDs are evaluated. Their A-CXL effects are investigated under both normal oxygen (21%) and hypoxic conditions, besides in vivo biocompatibility and possible functional mechanism of g-$C_3N_4$ QDs used as photosensitizer in the A-CXL process are presented. All of results suggest that the developed g-$C_3N_4$ QDs or RF@g-$C_3N_4$ QDs composite photosensitizers may serve as effective photosensitizer for A-CXL.

## Results and discussion
### Synthesis and characterization of g-$C_3N_4$ QDs

Figure 2a displayed a simple synthesis operation flow chart of g-$C_3N_4$ QDs. After synthesis, a light yellow transparent g-$C_3N_4$ QDs aqueous dispersion can be obtained, which exhibits excellent dispersibility in water even been lyophilized into g-$C_3N_4$ QDs powder (as shown in Supplementary Fig. 1). Additionally, different material characterizations were carried out to verify the successful synthesis of the g-$C_3N_4$ QDs. The results, presented in Fig. 2 and Supplementary Fig. 2–9, offer conclusive evidence. Specifically, the transmission electron microscope (TEM) and high-resolution TEM (HRTEM) images of the g-$C_3N_4$ QDs obtained at 350 °C (Fig. 2b, Supplementary Fig. 2) confirmed that the sample is mainly composed of nanoparticles of about 5–8 nm in diameter. A clear lattice fringe with a d-spacing of 0.45 nm can be observed in the HRTEM images (inserted figure in Fig. 2b), arising from the lattice fringe of the (002) plane of g-$C_3N_4$. The X-ray diffraction patterns obtained from g-$C_3N_4$ QDs samples obtained at different temperatures (Fig. 2c) were similar to that of conventional melamine-derived g-$C_3N_4$ bulk materials (BM), showing the same diffraction peaks at about 27.5°. The structure of the prepared g-$C_3N_4$ QDs obtained at different temperatures and g-$C_3N_4$ BM was also confirmed by FTIR spectra (Supplementary Fig. 3), which displayed a typical fingerprint of g-$C_3N_4$ materials in the range of 1200 to 1700 cm$^{-1}$, ascribed to υ(C-NH-C) and υ(C = N) stretching vibrations. An absorption peak located at 800 cm$^{-1}$ was attributed to the triazine ring breathing mode[33,34]. Two peaks at 3450 cm$^{-1}$ and 2170 cm$^{-1}$ were visible only in the spectra from g-$C_3N_4$ QDs and not in the g-$C_3N_4$ BM. The peak located at 3450 cm$^{-1}$ can be ascribed to the stretching vibrations of OH-groups incorporated into the poly(heptazine imide) structure after KOH/NaOH treatment. Additionally, the peak located at 2170 cm$^{-1}$ corresponds to the υ(C-N) vibrations[35]. X-ray photoelectron spectra (XPS) of the g-$C_3N_4$ QDs and g-$C_3N_4$ BM were also compared in this

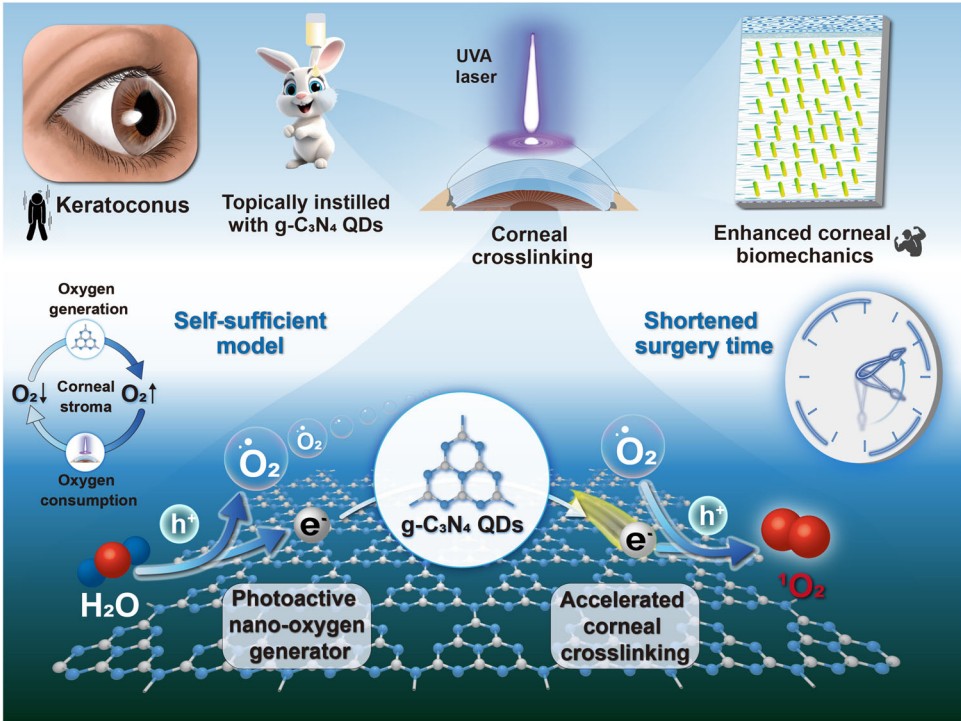

**Fig. 1 | Schematic diagram.** The A-CXL application of g-C₃N₄ QDs-based photosensitizer.

study and the results are shown in Fig. 2d–f and Supplementary Fig. 4. Compared with the scan spectrum of g-C₃N₄ BM (Supplementary Fig. 4a), the characterization peaks ascribed to Na and K can be observed in the scan spectrum of g-C₃N₄ QDs, indicating that Na and K remain in the production (Supplementary Fig. 4b, c), which can balance the charged equilibrium of the g-C₃N₄ QDs[35]. Then, the details of kinds of elements were further analyzed using a CasaXPS software. Compared with the C1s spectrum of g-C₃N₄ BM (Supplementary Fig. 4d), two more peaks located at 285.62 eV (peak 4, orange line) and 294.95 eV (peak 1, green line) can be observed in the C1s spectrum of g-C₃N₄ QDs except the different peak intensities (Fig. 2d). The peak located at 285.62 eV may be ascribed to the C-NH$_x$ and C-O⁻ groups, while the peak at the 294.95 eV should be ascribed to N-C = N bonds. These results indicate that the addition of the KOH and NaOH in the raw materials may change bonds around C atoms[35]. Comparing the N1s spectra of g-C₃N₄ BM (Supplementary Fig. 4e), one peak located at 400.89 eV (peak 2, green line) can be observed in the N1s spectrum of g-C₃N₄ QDs (Fig. 2e), which may be ascribed to C-NH$_x$. The difference of the O1s spectra further confirmed the obvious variation of the g-C₃N₄ QDs from g-C₃N₄ BM (Fig. 2f, Supplementary Fig. 4f), two more peaks located at 530.5 eV (peak 3, pink line) and 533.89 eV (peak 2, green line) can be observed in the O1s spectrum of g-C₃N₄ QDs, which may be ascribed to absorbed $O_2$, $H_2O$ and the existence of -OH[35].

Next, the luminescent properties of the g-C₃N₄ QDs were studied in detail. Diffuse reflection spectra shown in Fig. 2g revealed the difference of the absorbance range of the UV–visible light of the g-C₃N₄ QDs and g-C₃N₄ BM. Compared with g-C₃N₄ BM, g-C₃N₄ QDs has a wider absorbance range and stronger absorbance intensity, particularly within the 280–450 nm. Additionally, the influence of the annealing temperature on the UV–visible absorbance spectra of the g-C₃N₄ QDs was also studied. As exhibited in Supplementary Fig. 5, for the same concentration of g-C₃N₄ QDs, as the annealing temperature is increased, a slight blue-shift can be observed in the UV–visible absorbance spectra of the g-C₃N₄ QDs. When the annealing temperature was increased from 330 to 370 °C, the maximum of the UV–visible absorbance wavelength shifted from 363 to 361 nm. The concentration of

the g-C₃N₄ QDs aqueous dispersion can also influence the UV–visible absorbance ranges (Supplementary Fig. 6), an obvious blue–shift from 365 to 350 nm can be observed with the decrease of the g-C₃N₄ QDs concentration from 1 to 0.008 mg mL⁻¹, with the absorbance intensity reduced at the same time. All these results indicate that both the annealing temperature and the concentration of g-C₃N₄ QDs can influence the absorbance range of UV–visible light. Since 365 nm UVA light was used in CXL process, a slightly higher g-C₃N₄ QDs concentration was chosen in the subsequent experiments.

Fluorescence properties were also studied in detail. As shown in Fig. 2h and Supplementary Fig. 7, for the 500 µg mL⁻¹ g-C₃N₄ QDs aqueous dispersion monitored at 411 nm, an obvious excitation band with maximum at about 370 nm can be observed (Fig. 2h, blue line), while a strong emission band with a maximum at about 411 nm can be observed under excitation at 370 nm (Fig. 2h, red line), bright blue light can be observed for the g-C₃N₄ QDs aqueous dispersion when irradiated with 360 nm UVA light (Fig. 2a), confirming the successful synthesis of the g-C₃N₄ QDs. In addition, the impact of varying concentrations of the aqueous dispersion of g-C₃N₄ QDs on their luminescent properties was examined, and the findings are presented in Supplementary Fig. 7. As with UV–visible absorbance spectra, an obvious blue-shift can be observed with the decrease of the g-C₃N₄ QDs dispersion concentration, while the luminescent intensity increased with the dilution of the dispersion. A similar luminescent intensity variation can also be found in their emission spectra, with a maximum around 410 nm.

The Zeta potential of g-C₃N₄ QDs obtained at different temperatures was also studied, as shown in Supplementary Fig. 8, which compares the Zeta potential of g-C₃N₄ QDs obtained at 330 °C, 350 °C and 370 °C. It can be seen that the sample obtained at 350 °C has the lowest Zeta potential, indicating the sample obtained at this temperature may have more negatively charged functional groups on its surface.

The photocatalytic oxygen generation ability of g-C₃N₄ QDs obtained at different temperatures was also evaluated, and the results are shown in Fig. 2i and Supplementary Fig. 9. In comparison with the other samples, it was found that the photocatalytic oxygen generation

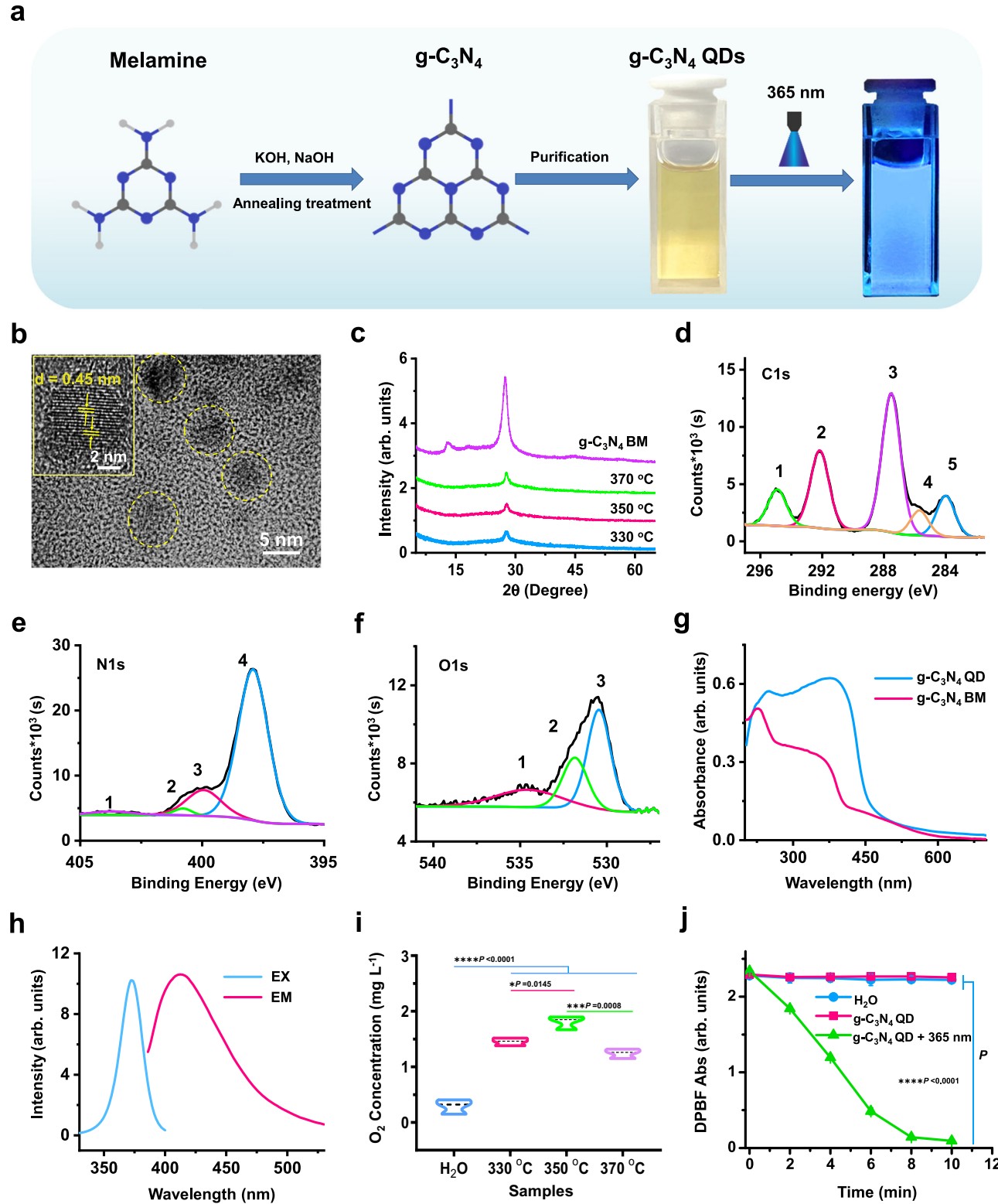

**Fig. 2 | Synthesis and characterization of the g-C₃N₄ QDs. a** A simple schematic diagram of the synthesis of g-C₃N₄ QDs. **b** TEM and HRTEM images of g-C₃N₄ QDs obtained at 350 °C. **c** X-ray diffraction patterns obtained from g-C₃N₄ BM and g-C₃N₄ QDs at different annealing temperatures. **d**–**f** C1s, N1s and O1s X-ray photoelectron spectra of g-C₃N₄ QDs obtained at 350 °C using a CasaXPS analysis software processed. **g** Diffuse reflection spectra of the g-C₃N₄ QDs and g-C₃N₄ BM. **h** Excitation (EX) and emission (EM) spectra of g-C₃N₄ QDs obtained at 350 °C (0.5 mg mL⁻¹). **i** O₂ generation ability of g-C₃N₄ QDs obtained at different temperatures (mean ± SD, $n = 3$, one-way ANOVA multiple comparison test, the

absence of a *P*-value indicates no statistical difference between the groups ($p > 0.05$)). **j** Singlet oxygen generation capability of g-C₃N₄ QDs obtained at 350 °C using DPBF as chemical probe by detecting its absorption intensity at 410 nm. Blue line: H₂O without irradiation with 365 nm UVA light; Pink line: g-C₃N₄ QDs aqueous dispersion without irradiation with 365 nm UVA light; Green line: g-C₃N₄ QDs aqueous dispersion with irradiation with 365 nm UVA light (mean ± SD, $n = 4$, two-way ANOVA multiple comparison test, the absence of a *P*-value indicates no statistical difference between the groups ($p > 0.05$)).

ability was greatest in g-C$_3$N$_4$ QDs obtained at 350 °C, after 15 min irradiation using 365 nm UVA light, the concentration of oxygen in the aqueous dispersion increased by about 2 mg L$^{-1}$, indicating the excellent photocatalytic oxygen generation ability of that sample. Besides, water-splitting reaction system was further used to evaluate the photocatalytic O$_2$ evolution performance of g-C$_3$N$_4$ QDs obtained at 350 °C under UV-vis light illumination using 10 vol% triethanolamine as a sacrificial agent. As illustrated in Supplementary Fig. 9, the obtained g-C$_3$N$_4$ QDs samples exhibit substantial enhancement in O$_2$ evolution performances, the O$_2$ evolution rate was gradually increased from 0.008 mmol g$^{-1}$ h$^{-1}$ to 0.02 mmol g$^{-1}$ h$^{-1}$, indicating excellent photocatalytic O$_2$ generation performance.

Furthermore, the singlet oxygen generation ability of the g-C$_3$N$_4$ QDs obtained 350 °C was also studied. In Fig. 2j, a chemical probe, known as DPBF, was employed to confirm the generation ability of singlet oxygen by monitoring absorption intensity at 410 nm via UV−visible spectroscopy. Compared with the non-irradiated H$_2$O and g-C$_3$N$_4$ QDs aqueous dispersion (Fig. 2j, blue line and pink line), the g-C$_3$N$_4$ QDs aqueous dispersion with the irradiation of 365 nm UVA light showed a dramatic decrease with time in the absorption intensity of DPBF at 410 nm (Fig. 2j, green line), indicating the formation of a large amount of singlet oxygen during the 10-minute irradiation process. Electron spin resonance (ESR) method was also used to test the production of singlet oxygen of g-C$_3$N$_4$ QDs aqueous dispersion under the irradiation of UVA light. As shown in Supplementary Fig. 10, the content of singlet oxygen increased with the prolonging of the irradiation, further confirming the excellent singlet oxygen generation capability of the obtained g-C$_3$N$_4$ QDs under UVA light irradiation.

Besides, the stability of the synthesized g-C$_3$N$_4$ QDs dispersed in various solutions, such as deionized water (H$_2$O), PBS, 10% fetal bovine serum(FBS) aqueous solution (10% FBS), and 100% FBS, was studied. Since the samples can be lyophilized into g-C$_3$N$_4$ QDs powder and preserved for extended periods, only the state changes of the g-C$_3$N$_4$ QDs dispersed in these various solutions at 4 °C within 10-days timeframe were provided in this study. Photographs were captured from both the front view (FV) and the top view (TV) of the dispersions, with the side of the dispersions irradiated by laser to exhibit the Tyndall effect. As depicted in Supplementary Fig. 11, the g-C$_3$N$_4$ QDs dispersed in the four solvents retained their transparency and clarity over time, indicating that the synthesized g-C$_3$N$_4$ QDs maintain good dispersibility and stability in different solutions. Particle size test results of the g-C$_3$N$_4$ QDs dispersed in these solutions further confirmed their good stability and dispersibility. However, the hydrated particle size of the samples varies in different dispersions, possibly due to different solvent effects, as illustrated in Supplementary Fig. 12.

## In vitro biological compatibility evaluation of g-C$_3$N$_4$ QDs

Prior to performing in vivo studies, the biological safety of different concentrations of g-C$_3$N$_4$ QDs (up to 800 μg mL$^{-1}$) were studied in vitro. CCK-8 assay, Calcein-AM/PI double staining, and cell apoptosis experiments were carried out. Human corneal epithelial cells (HCEC cells), human retinal microvascular endothelial cells (hRMEC), and retinal pigment epithelium cells (RPE-19) were used in the CCK-8 assay. After 24 h of incubation with g-C$_3$N$_4$ QDs at different concentrations, the survival rate of three types of cells was measured, and the results are shown in Fig. 3a−c. Compared with the control group (no g-C$_3$N$_4$ QDs doped), the survival rates of the different types of cells remained higher than 100% even when the concentration of g-C$_3$N$_4$ QDs was increased to 800 μg mL$^{-1}$, demonstrating that g-C$_3$N$_4$ QDs have excellent biocompatibility, and the application of g-C$_3$N$_4$ QDs in certain concentration can cause a proliferation of certain type of cell. Next, Calcein-AM/PI double staining was used to evaluate the biological safety of the g-C$_3$N$_4$ QDs. As illustrated in Fig. 3d, the quantity of living cells (green fluorescence) in the visual field in the Calcein-AM groups is similar to that in the Merge groups, only several dead cells (red fluorescence) in the

visual field in PI groups can be found even when the concentration of g-C$_3$N$_4$ QDs is increased to 400 μg mL$^{-1}$, demonstrating that the g-C$_3$N$_4$ QDs had excellent cell safety. Meanwhile, the Annexin V-FITC/PI cell apoptosis assay kit was further employed to evaluate the proportions of living, early apoptotic, late apoptotic, and necrotic HCEC cells after 24 h incubation with different concentrations of g-C$_3$N$_4$ QDs. The corresponding gating strategy for this analysis is presented in Supplementary Fig. 13. As illustrated in Fig. 3e, it is noteworthy that even after incubation with 400 μg mL$^{-1}$ g-C$_3$N$_4$ QDs for 24 h, the percentage of living cells remained around 90%. This demonstrated that the g-C$_3$N$_4$ QDs had excellent cell safety, and was consistent with the results of CCK-8 and Calcein-AM/PI double staining.

## A-CXL evaluation of g-C$_3$N$_4$ QDs

Prior to the in vivo A-CXL evaluation, a comparison was made between the transparency of the deepithelialized cornea soaked with g-C$_3$N$_4$ QDs aqueous dispersion and that of an untreated cornea. After being presoaked for 30 min using a 2.5 mg mL$^{-1}$ concentration of g-C$_3$N$_4$ QDs aqueous dispersion, no discernible differences can be observed in cornea transparent. Specifically, the treated cornea retained 100% transmittance when exposed to a controlled light wavelength of 510 nm (Supplementary Fig. 14a), similar to that of the untreated cornea. Additionally, photographs documenting the treated corneas submerged in PBS solution were collected and presented in Supplementary Fig. 14b. The clear visibility of the letter "E" indicated the excellent transparency of the treated cornea.

Then, the A-CXL evaluation of g-C$_3$N$_4$ QDs was carried out, and the findings are presented in Figs. 4, 5, 6, 9 and Supplementary Fig. 15−17. First, the influence of the different UVA intensity on the CXL effect of g-C$_3$N$_4$ QDs was studied. A simple operation flow chart of A-CXL evaluation is shown in Fig. 4a. After being presoaked for 30 min using 2.5 mg mL$^{-1}$ g-C$_3$N$_4$ QDs, the cornea was irradiated using 365 nm continuous UVA light with an 8 mm facula. The intensity of the continuous UVA laser was changed whilst keeping the total energy at 5.4 J. Five groups were selected: PBS (without irradiation), 3 mW cm$^{-2}$ (g-C$_3$N$_4$ QDs, 3 mW cm$^{-2}$, 30 min), 6 mW cm$^{-2}$ (g-C$_3$N$_4$ QDs, 6 mW cm$^{-2}$, 15 min), 9 mW cm$^{-2}$ (g-C$_3$N$_4$ QDs, 9 mW cm$^{-2}$, 10 min) and 18 mW cm$^{-2}$ (g-C$_3$N$_4$ QDs, 18 mW cm$^{-2}$, 5 min). After that, enzymatic digestion of collagen was performed to assess the CXL effect, and the results are shown in Figs. 4b−d. Photographs of the corneal tissue residues after enzymatic digestion of CXL are presented in Fig. 4b. Compared with the 6 h it took for the corneal tissue to dissolve completely in the PBS group, the application of g-C$_3$N$_4$ QDs prolonged the dissolution time to more than 8 h for the 3 mW cm$^{-2}$ group (3 mW cm$^{-2}$, 30 min exposure), indicating a weak enzymatic resistance at this UVA intensity. When the UVA intensity was increased to 6 mW cm$^{-2}$ with the irradiation time decreased to 15 min, even after 48 h digestion, the corneal residues had a disc area about 25% of their initial state, indicating the dramatic enhancement of the CXL effect. Further increasing the UVA intensity resulted in little change in the rate of enzymatic digestion and the disc area. Combined with the quantitative analysis shown in Fig. 4c, the time for the residual cornea in the different groups to reach equilibrium (i.e. no further change in the residual cornea disc area) was different. It took about 32 h for the corneal residues in the 9 mW cm$^{-2}$ and 18 mW cm$^{-2}$ groups to reach balance, with minimal changes in the disc area despite prolonged the soaking time, while the disc area for the corneal residues in the 6 mW cm$^{-2}$ group went on decreasing even after 48 h. The dry weights of the residual corneas after being soaked in 3 mL 0.2% Collagenase Type 2 solution for 48 h were further measured, and the results are shown in Fig. 4d. The dry weight of the residual cornea in the 9 mW cm$^{-2}$ group was higher than that in the 6 mW cm$^{-2}$ and 18 mW cm$^{-2}$ groups with significant difference, with approximately 0.9 mg corneal tissue was left, indicating the excellent A-CXL effect using a 9 mW cm$^{-2}$ UVA laser for 10 min. Furthermore, scanning electron microscope (SEM) analysis was conducted to investigate the

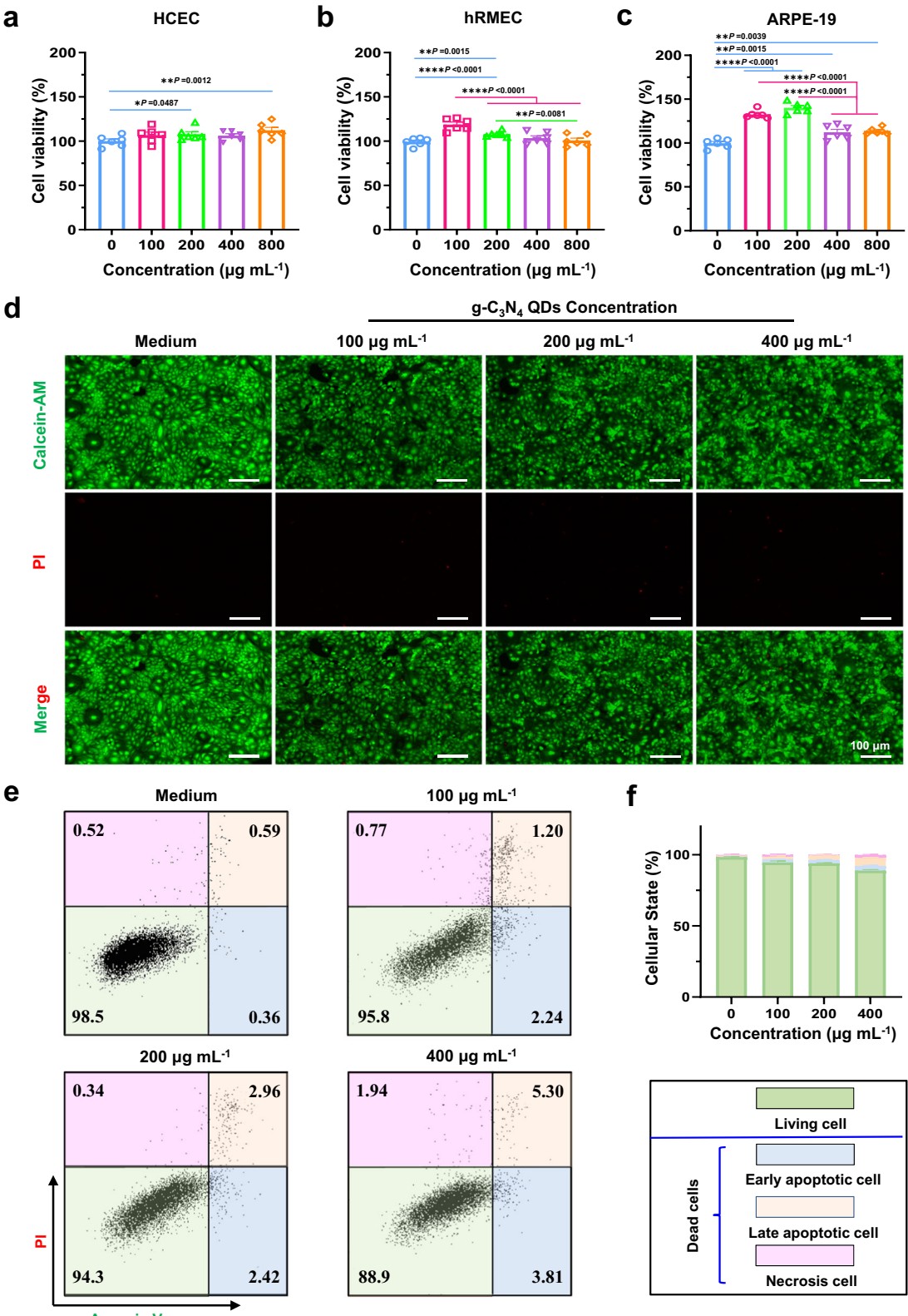

alterations in the corneal structure after A-CXL in the 9 mW cm$^{-2}$ group, with comparisons made to the PBS group. As illustrated in Fig. 4e, the cornea treated with g-C$_3$N$_4$ QDs exhibited an increased density of collagen fibrils and reduced interstitial spaces between collagen layers. These findings serve as conclusive evidence of the effective application of A-CXL using g-C$_3$N$_4$ QDs as a photosensitizer under UVA irradiation.

The above results also suggest that it is important to choose the correct UVA intensity and irradiation time to enhance the A-CXL effect when using a g-C$_3$N$_4$ QDs aqueous dispersion as the CXL photosensitizer. Due to the dual function, g-C$_3$N$_4$ QDs play in the A-CXL process, both as a catalyst for photocatalytic oxygen production and as a photosensitizer for CXL, the intensity of UVA used in the C-CXL protocol (3 mW cm$^{-2}$ for 30 min) is not enough to support these dual

**Fig. 3 | In vitro biocompatible evaluation of g-C$_3$N$_4$ QDs. a** Cell viability of HCEC cells incubated with g-C$_3$N$_4$ QDs at different concentrations (biologically independent samples, two-way ANOVA multiple comparison test, mean ± SD, $n = 6$, the absence of a $P$-value indicates no statistical difference between the groups ($p > 0.05$)). **b** Cell viability of hRMEC cells incubated with g-C$_3$N$_4$ QDs at different concentrations (biologically independent samples, two-way ANOVA multiple comparison test, mean ± SD, $n = 6$, the absence of a $P$-value indicates no statistical difference between the groups ($p > 0.05$)). **c** Cell viability of ARPE−19 cells incubated with g-C$_3$N$_4$ QDs at different concentrations (biologically independent samples, two-way ANOVA multiple comparison test, mean ± SD, $n = 6$, the absence of a $P$-value indicates no statistical difference between the groups ($p > 0.05$)). **d** Calcein-AM/PI double staining of HCEC cells incubated with g-C$_3$N$_4$ QDs at

different concentrations (The experiment was repeated three times independently with similar results). **e, f** Flow-cytometric analysis of HCEC cells incubated with g-C$_3$N$_4$ QDs at different concentrations. (biologically independent samples, two-way ANOVA multiple comparison test, mean ± SD, $n = 3$, living cell (0 vs 100, $p < 0.0001$; 0 vs 200, $p < 0.0001$; 0 vs 400, $p < 0.0001$; 100 vs 400, $p < 0.0001$; 200 vs 400, $p < 0.0001$), Early apoptotic cell (0 vs 100, $p < 0.0001$; 0 vs 200, $p < 0.0001$; 0 vs 400, $p < 0.0001$); Late Apoptotic Cell (0 vs 100, $p = 0.0027$; 0 vs 200, $p < 0.0001$; 0 vs 400, $p < 0.0001$; 100 vs 200, $p = 0.0044$; 100 vs 400, $p < 0.0001$; 200 vs 400, $p < 0.0001$), Necrosis Cell (0 vs 400, $p = 0.0019$; 200 vs 400, $p = 0.0009$), the absence of a $P$-value indicates no statistical difference between the groups ($p > 0.05$)).

---

functions of the g-C$_3$N$_4$ QDs (Fig. 4b, c). Our results indicate that this can be improved by using a 6 mW cm$^{-2}$ UVA laser for 15 min (Fig. 4c), but the effect cannot be increased even when the irradiation time is prolonged to 30 min (total energy 10.8 J) (Supplementary Fig. 15). When the UVA intensity was increased from 3 to 9 mW cm$^{-2}$, the A-CXL effect was enhanced obviously with significant difference (Fig. 4d). As before, prolonging the irradiation time did not significantly improve the A-CXL effect (Supplementary Fig. 16). However, further increasing the UVA intensity to 18 mW cm$^{-2}$ resulted in a reduction of the A-CXL effect. This could be due to the short irradiation time (5 min), which results in an imbalance between the O$_2$ generating capabilities of the g-C$_3$N$_4$ QDs and the more rapid consumption of O$_2$ at higher UVA intensities. The oxygen consumption in the cornea increases sharply with the increase of UVA intensity, while the oxygen in the stroma cannot be replenished in this time, resulting a reduction of the A-CXL effect[36]. These phenomena prove the importance of using the correct combination of UVA intensity and exposure time to maximize the dual function of g-C$_3$N$_4$ QDs in the CXL process, and 9 mW cm$^{-2}$ with 10 min irradiation was used in the following CXL evaluation of different samples.

Then, the influence of the different sample on the A-CXL effect is further studied, a simple operation flow chart of A-CXL evaluation of different samples is shown in Fig. 5a. After being presoaked for 30 min with different samples including PBS, RF, g-C$_3$N$_4$ QDs, and RF@g-C$_3$N$_4$ QDs, 365 nm of continuous UVA with 8 mm facula and 9 mW cm$^{-2}$ intensity was used to irradiate the cornea, the irradiation time was controlled at 10 min. Meanwhile, 3 mW cm$^{-2}$ (30 min) UVA was also applied on RF 3 mW cm$^{-2}$ group to mimic conventional CXL protocol. After that, in vitro enzymatic digestion of collagen, stress-strain measurements of cornea, as well as in vivo biomechanical evaluation using Corvis® ST and Optovue RTVue OCT were conducted to assess the A-CXL effect of different samples. Figure 5b displays representative images of the corneal tissue residues in each treatment group with different samples during the process of enzymatic digestion with collagenase. The corneal disks in the PBS group, took approximately 6 h to dissolve completely. The rate of enzymatic digestion was far slower in the g-C$_3$N$_4$ QDs, Rf@g-C$_3$N$_4$ QDs and Rf CXL groups, and in all cases corneal tissue was still present after 48 h enzymatic digestion. However, the time at which the digestion of the residual cornea stopped was different (Fig. 5c), it took about 32 h for the g-C$_3$N$_4$ QDs group and RF@g-C$_3$N$_4$ QDs group to stop being digested, after which, further extension of soaking time had minimal impact on the area of the residues. On the other hand, longer time were needed for the corneal residues in the two RF groups to stop being digested. Furthermore, it can also be seen in Fig. 5c that the disc areas of the residual corneas in the four groups were also different. The disc areas of the residual corneas in the g-C$_3$N$_4$ QDs and RF@g-C$_3$N$_4$ QDs groups were both about 28% of their initial state, which is much higher than the RF 9 mW cm$^{-2}$ group where the area was only about 13% of the initial state, and comparable to that of conventional group (RF 3 mW cm$^{-2}$). After soaking for 48 h, the dry weights of the residual corneas were further measured and shown in Fig. 5d. The results indicated that the dry weight of remains in the g-C$_3$N$_4$ QDs and RF@g-C$_3$N$_4$ QDs groups was

slightly higher than that of those in the RF groups (RF 3 mW cm$^{-2}$ and RF 9 mW cm$^{-2}$).

Stress-strain measurements of cornea strips were also conducted to assess the A-CXL effect in the different groups. As shown in Fig. 5e, f, the ultimate stress and the Young's modulus of the corneas in the g-C$_3$N$_4$ QDs and RF@g-C$_3$N$_4$ QDs groups were much higher compared to the PBS and RF 9 mW cm$^{-2}$ groups under the same conditions, on par with to the conventional RF CXL protocol (3 mW cm$^{-2}$, 30 min) with no significant difference, indicating excellent strengthening of the cornea after A-CXL treatments when using g-C$_3$N$_4$ QDs or RF@g-C$_3$N$_4$ QDs composite as a photosensitizer. While the comparable enzymatic digestion of collagen and stress-strain measurements between g-C$_3$N$_4$ QDs and RF@g-C$_3$N$_4$ QDs groups reveal that the addition of RF in the g-C$_3$N$_4$ QDs aqueous dispersion has weak enhancement of the CXL effect.

Concurrently, an evaluation of the in vivo biomechanical properties of the cornea was conducted following CXL treatment with various photosensitizers at different time points (1, 3, 5, 7, 14, 21 days) using Corvis® ST. Key corneal biomechanical parameters, including the first applanation length (A1L), the second applanation velocity (A2V), the radius at highest concavity (HC-R), and deformation amplitude at highest concavity (HC-DA), were selected and compared with those of normal corneas (PBS group). Among the parameters measured by Corvis® ST, the flattening speed serves as a proxy for the overall stiffness and resistance of the cornea. As corneal stiffness increases, so does its resistance, resulting in a reduced compression speed in response to airflow. The maximum deformation amplitude of the cornea, on the other hand, reflects its softness. The harder the cornea, the smaller deformation amplitude. As shown in Fig. 6b−e, corneal edema, resulting from the removal of corneal epithelium, can influence the data collected within the initial few days, leading to some discrepancies when compared to other groups. Nonetheless, statistically significant differences can be observed in the A1L values of the corneas in the g-C$_3$N$_4$ QDs group when compared to both the PBS and RF 9 mW cm$^{-2}$ groups under identical conditions. These values are comparable to those obtained using the conventional RF CXL protocol (RF 3 mW cm$^{-2}$) with no significant difference, indicating a positive A-CXL effect when using g-C$_3$N$_4$ QDs as a photosensitizer (Fig. 6b). Furthermore, the observed slower A2V values (Fig. 6c), the increase in HC-R (Fig. 6d), and the smaller HC-DA values (Fig. 6e) observed in the corneas of the g-C$_3$N$_4$ QDs group suggest an improvement in corneal biomechanical properties following A-CXL using g-C$_3$N$_4$ QDs as a photosensitizer. Collectively, these findings underscore the potential of g-C$_3$N$_4$QDs as effective photosensitizers in A-CXL treatments aimed at enhancing corneal biomechanical properties.

Additionally, measurements of central corneal thickness (CCT) were also collected from the corneas in various groups at different time points (21, 30 days) following CXL treatment. These measurements are presented in Figs. 6f, g and Supplementary Fig. 17. As time progressed following CXL, corneal edema gradually abated. Notably, The CCT values for the corneas in both the g-C$_3$N$_4$ QDs and RF

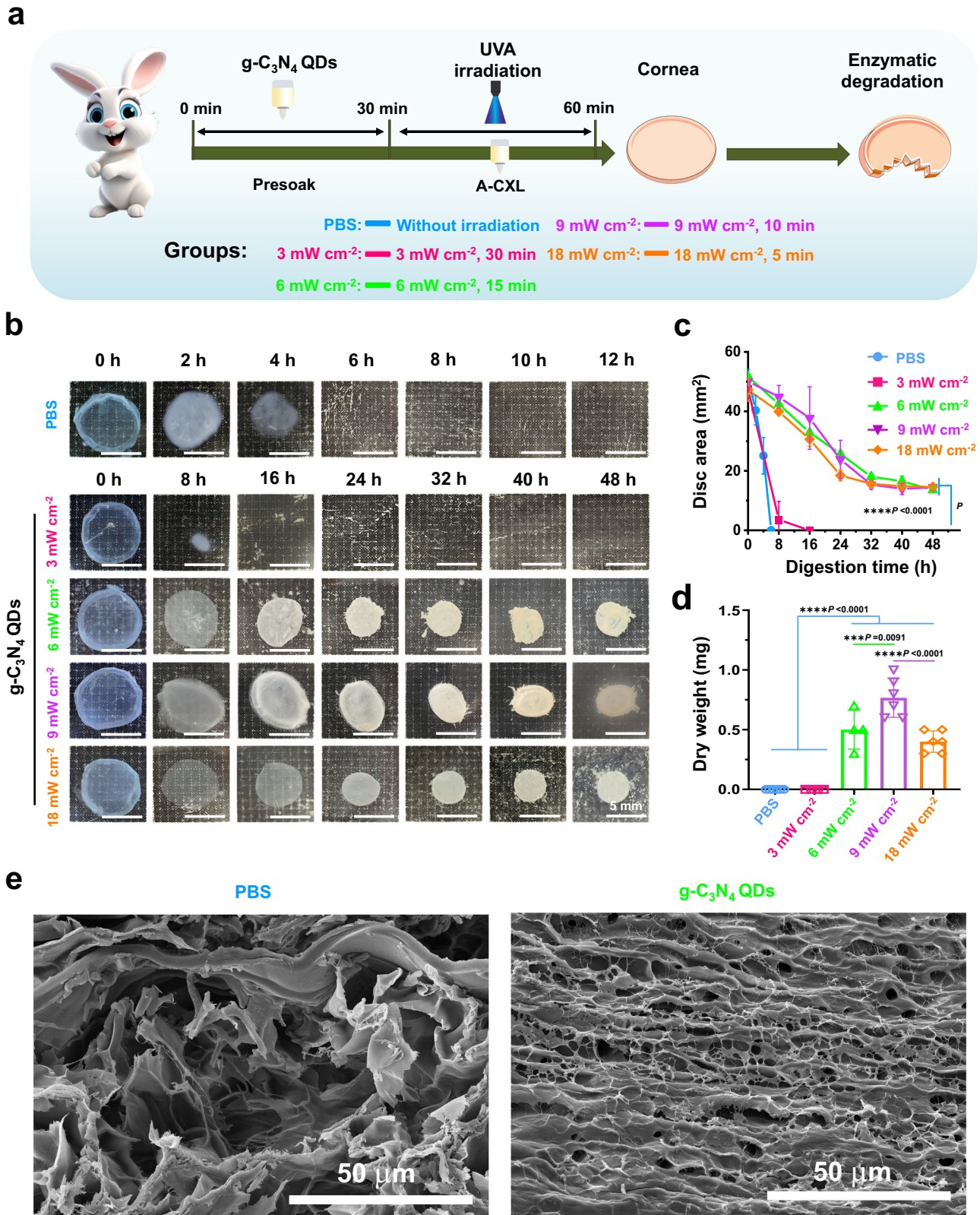

**Fig. 4 | The influence of the variant UVA intensity on A-CXL efficacy of C₃N₄ QDs. a** Schematic of a simple operation flow chart of A-CXL evaluation including variable experimental condition. **b** Photos depicting the digestion behavior of the cornea in the presence of collagenase II over time. **c** Quantitative analysis of the undigested corneal disc area over time (mean ± SD, PBS group, $n = 6$; 3 mW cm⁻² and 6 mW cm⁻² groups, $n = 4$; 9 mW cm⁻² and 18 mW cm⁻² groups, $n = 6$; two-way ANOVA multiple comparison test, the absence of a $P$-value indicates no

statistical difference between the groups ($p > 0.05$)). **d** The dry weight of the residual cornea (mean ± SD, PBS group, $n = 6$; 3 mW cm⁻² and 6 mW cm⁻² groups, $n = 4$; 9 mW cm⁻² and 18 mW cm⁻² groups, $n = 6$; two-way ANOVA multiple comparison test, the absence of a $P$-value indicates no statistical difference between the groups ($p > 0.05$)). **e** SEM images of the corneal issues in PBS and g-C₃N₄ QDs groups (The experiment was repeated three times independently with similar results).

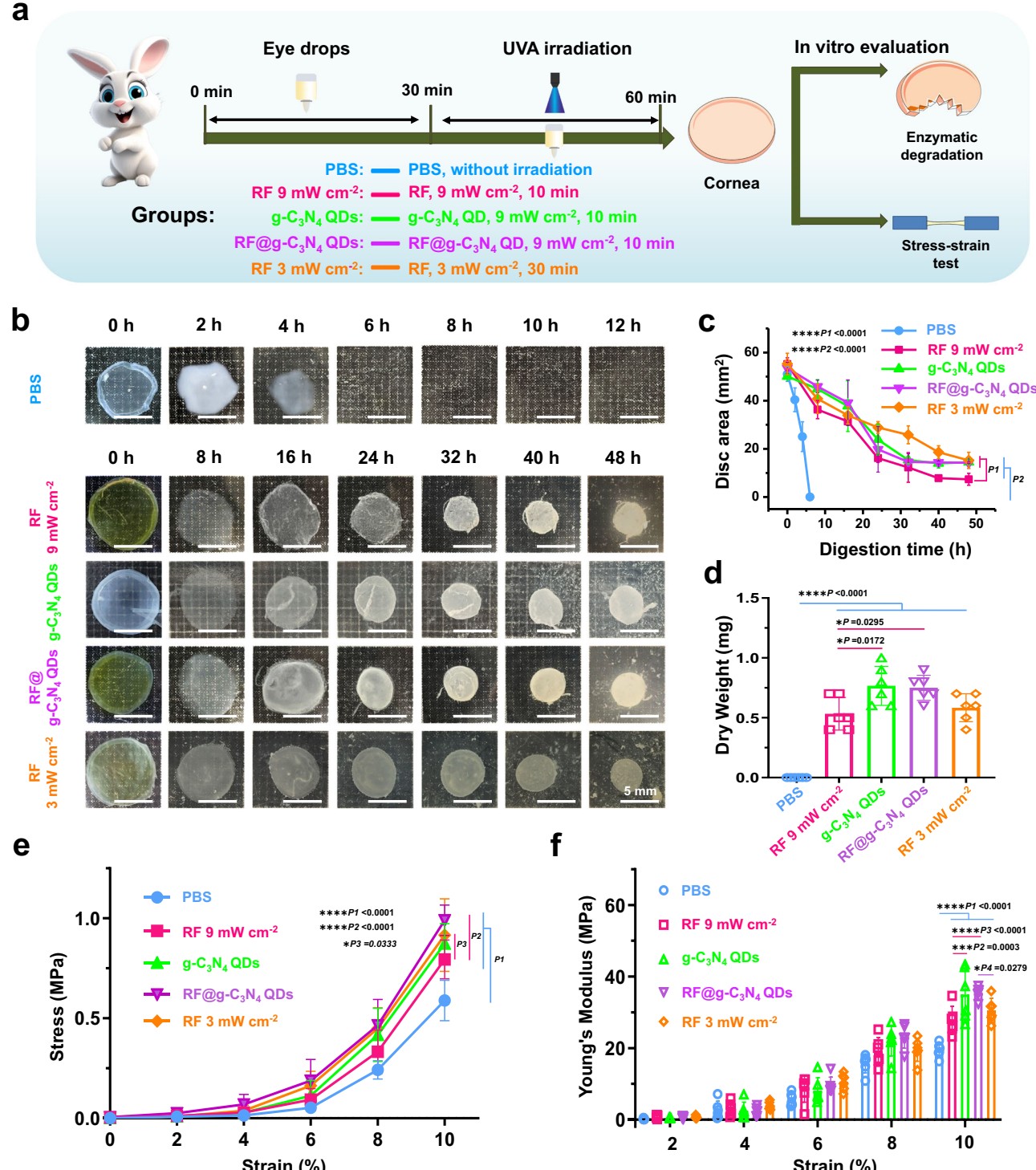

**Fig. 5 | The comparison of corneal biomechanical property with different treatment. a** Schematic of a simple operation flow chart of A-CXL evaluation. **b** depicting the digestion behavior of the cornea in the presence of collagenase II over time. The cornea in PBS group is with no any treatment, the corneas in RF 9 mW cm⁻², g-C₃N₄ QD and RF@g-C₃N₄ groups was irradiated with 9 mW cm⁻² UVA laser for 10 min UVA, while the cornea in RF 3 mW cm⁻² group was irradiated with 3 mW cm⁻² UVA laser for 30 min UVA. **c** Quantitative analysis of the disc area over time (mean ± SD, $n = 6$, two-way ANOVA multiple comparison test, the absence of a

$P$-value indicates no statistical difference between the groups ($p > 0.05$)). **d** The dry weight of residual cornea after soaking for 48 h (mean ± SD, $n = 6$, two-way ANOVA multiple comparison test, the absence of a $P$-value indicates no statistical difference between the groups ($p > 0.05$)). **e**, **f** The biomechanical properties of corneas after A-CXL and C-CXL (mean ± SD, $n = 6$, two-way ANOVA multiple comparison test, the absence of a $P$-value indicates no statistical difference between the groups ($p > 0.05$)).

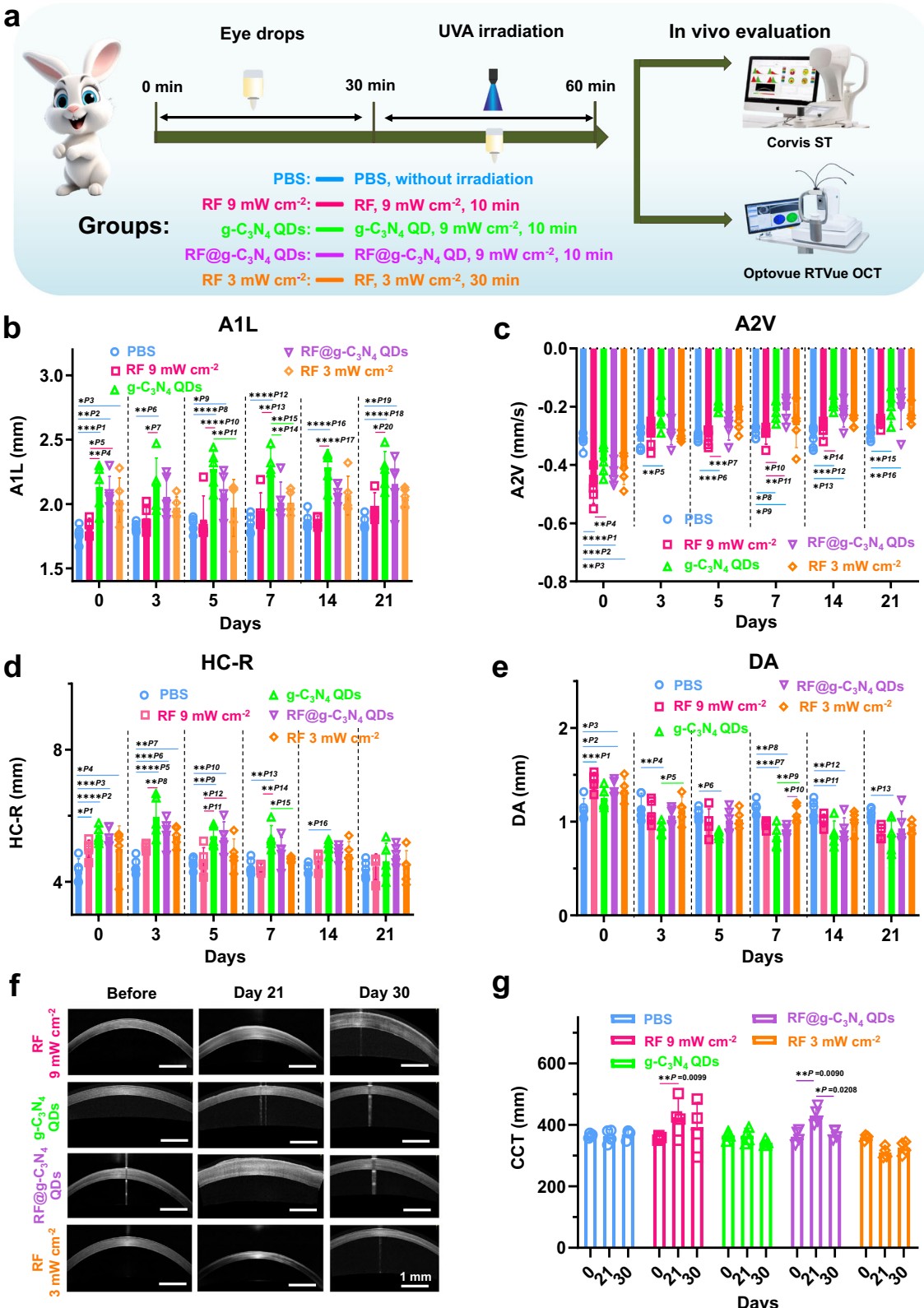

3 mW cm⁻² groups approximated those recorded prior to A-CXL when the observation period was extended to 21 days. With a further prolongation of the observation period, minimal reductions in CCT values were observed (Fig. 6g and Supplementary Fig. 17), indicating a faster recovery and healing effect of these two protocols. Nevertheless, it is worth mentioning that corneal edema persisted in a significant manner in the corneas of both the RF 9 mW cm⁻² and RF@g-$C_3N_4$ QDs groups

even after 21 days. When the observation period was extended to 30 days, a distinct difference emerged between these two groups, unlike the RF 9 mW cm⁻² group, where the CCT values remained elevated compared to pre-CXL levels, the RF@g-$C_3N_4$ QDs group exhibited CCT values comparable to those recorded before CXL. Thus, the existence of g-$C_3N_4$ QDs may facilitate the recovery and healing of cornea.

**Fig. 6 | The corneal structure variation after A-CXL treatment. a** Schematic of a simple operation flow chart of A-CXL evaluation using Corvis® ST and Optovue RTVue OCT. **b** Comparison of A1L (mean ± SD, $n = 5$, (for RF 9 mW cm$^{-2}$ group, $n = 4$, one rabbit died on Day 10), two-way ANOVA multiple comparison test, $P1 = 0.0003$, $P2 = 0.0018$, $P3 = 0.0196$ $P4 = 0.0061$, $P5 = 0.0257$, $P6 = 0.0014$, $P7 = 0.0210$, $P8 < 0.0001$, $P9 = 0.0406$, $P10 < 0.0001$, $P11 = 0.0048$, $P12 < 0.0001$, $P13 = 0.0033$, $P14 = 0.0381$, $P15 = 0.0335$, $P16 < 0.0001$, $P17 < 0.0001$, $P18 < 0.0001$, $P19 = 0.0057$, $P20 = 0.0153$, the absence of a $P$-value indicates no statistical difference between the groups ($p > 0.05$)). **c** Comparison of A2V (mean ± SD, $n = 5$, (for RF 9 mW cm$^{-2}$ group, $n = 4$, one rabbit died on Day 10), two-way ANOVA multiple comparison test, $P1 < 0.0001$, $P2 = 0.0001$, $P3 = 0.0024$, $P4 = 0.0052$. $P5 = 0.0089$, $P6 = 0.0007$, $P7 = 0.0009$, $P8 = 0.0355$, $P9 = 0.0229$, $P10 = 0.0144$, $P11 = 0.0089$, $P12 = 0.0001$, $P13 = 0.0113$, $P14 = 0.0121$, $P15 = 0.0024$, $P16 = 0.0054$, the absence of a $P$-value indicates no statistical difference between the groups ($p > 0.05$)). **d** Comparison of HC-R (mean ± SD, $n = 5$, (for RF 9 mW cm$^{-2}$ group, $n = 4$, one rabbit died on Day 10), two-way ANOVA multiple comparison test, $P1 = 0.0368$, $P2 < 0.0001$, $P3 = 0.0002$, $P4 = 0.0386$. $P5 < 0.0001$, $P6 = 0.0002$, $P7 = 0.0016$, $P8 = 0.0014$, $P9 = 0.0048$, $P10 = 0.0076$, $P11 = 0.0131$, $P12 = 0.0199$, $P13 = 0.0026$, $P14 = 0.0012$, $P15 = 0.0377$, $P16 = 0.0455$, the absence of a $P$-value indicates no statistical difference between the groups ($p > 0.05$)). **e** Comparison of HC-DA (mean ± SD, $n = 5$, (for RF 9 mW cm$^{-2}$ group, $n = 4$, one rabbit died on Day 10), two-way ANOVA multiple comparison test, $P1 = 0.0003$, $P2 < 0.0190$, $P3 = 0.0310$, $P4 = 0.0087$, $P5 = 0.0493$, $P6 = 0.0135$, $P7 = 0.000$, $P8 = 0.0019$, $P9 = 0.0066$, $P10 = 0.0493$, $P11 = 0.0021$, $P12 = 0.0079$, $P13 = 0.0190$, the absence of a $P$-value indicates no statistical difference between the groups ($p > 0.05$)). **f** Photos of OCT images. **g** Comparison of the CCT (mean ± SD, $n = 4$, two-way ANOVA multiple comparison test, the absence of a $P$-value indicates no statistical difference between the groups ($p > 0.05$)).

## The eye biocompatibility of g-C₃N₄ QDs

The biocompatibility of g-C$_3$N$_4$ QDs in the cornea during various CXL treatments was also studied. Prior to CXL treatment, an ocular surface irritation evaluation was conducted on rabbits. Utilizing slit lamp microscopy, the integrity of the corneal epithelium was carefully observed. Photographic documentation was collected at 6 h and 24 h post-administration. As illustrated in Fig. 7a, when compared to the PBS control group, no staining of the corneal epithelium or epithelial defect was detected following fluorescein sodium staining. Meanwhile, there were no apparent symptoms of anterior ganglia irritation. Specifically, the cornea remained transparency without turbidity, the conjunctiva showed no signs of congestion or edema, there was no iris involvement, and no abnormal secretions were present within the 24-hour observation period. These findings indicate that the obtained g-C$_3$N$_4$ QDs dispersion exhibits weak ocular surface irritation.

After the application of A-CXL treatment, corneal recovery was initially monitored using a slit-lamp microscope, with the RF 9 mW cm$^{-2}$ group serving as a control. As shown in Supplementary Fig. 18, during the initial days following the treatment, both groups exhibited notable signs of anterior ganglion irritation, corneal opacity, conjunctival hyperemia and edema, iris involvement, and abnormal secretions. Notably, these symptoms appeared to be more pronounced in the RF 9 mW cm$^{-2}$ group. These manifestations were primarily attributed to the removal of the corneal epithelium and gradually subsided over time. By the 7th day post-treatment, a significant recovery of ocular tissues was observed for g-C$_3$N$_4$ QDs group. By the 14th day, the eye had essentially reverted to its pre-surgical state. As time progressed, minimal to no discernible differences can be observed in the cornea upon slit-lamp examination.

Meanwhile, the corneal endothelium of rabbits subjected to various treatments was evaluated using a specular microscope (EM-4000). Data on the endothelial cell count from different groups post-CXL treatment were gathered and analyzed. The results, as presented in Fig. 7b, revealed a substantial decrease in the number of endothelial cells in the RF 3 mW cm$^{-2}$ group following CXL (with significant difference of "$p < 0.0001$"). In contrast, the A-CXL groups, particularly the g-C$_3$N$_4$ QDs group, demonstrated only minor reductions in endothelial cell count (with significant difference of "$p = 0.0382$"). These results suggest that A-CXL treatment, especially when utilizing g-C$_3$N$_4$ QDs as a photosensitizer, offers greater safety in terms of endothelial cell preservation (The observed overall decline in data across every experimental group might be attributed to the change of the rabbits' growth environment, the error margin of specular microscope, and potentially other variables). This conclusion is further corroborated by the images shown in Supplementary Fig. 19, which exhibit well-maintained endothelial cell morphology and a smaller reduction in cell count in the treated eye compared to the untreated eye.

Moreover, the biocompatibility of g-C$_3$N$_4$ QDs on the cornea in vivo without and with A-CXL were also compared through histopathologic observations. Hematoxylin-Eosin staining of the cornea, Alizarin Red S and Trypan Blue staining of the endothelium, as well as TUNEL staining were also carried out, and the results are shown in Figs. 7c–e. The Hematoxylin-Eosin staining results (Fig. 7c) showed no obvious differences between the PBS group and the other irradiated and non-irradiated treatment groups, all cells in the stromal layer evenly distributed, and no apparent cell damage was observed. This indicates the excellent biocompatibility of g-C$_3$N$_4$ QDs, even upon exposure to 9 mW cm$^{-2}$ UVA light for 10 min. Endothelium staining (ES) using Alizarin Red S and Trypan Blue were also carried out for the different groups without A-CXL and with A-CXL. As shown in Fig. 7d, similar to the PBS and RF 9 mW cm$^{-2}$ groups, no blue cells were detected in the g-C$_3$N$_4$ QDs group. This indicates that all cells remain healthy, even after undergoing A-CXL. The preserved endothelial structure further underscores the healthy state of cells in the g-C$_3$N$_4$ QDs group without and with A-CXL. This conclusion was further confirmed by the quantitative measurements from the ES photos (insert in Fig. 7d). TUNEL staining of the cornea was also conducted to assay the apoptosis effects of the synthesized g-C$_3$N$_4$ QDs. As Fig. 7e illustrates, with the standard positive and negative group displayed in Supplementary Fig. 20. No signs of apoptosis (indicated by green color) can be observed in the images of the g-C$_3$N$_4$ QDs group without A-CXL, confirming the exceptional biocompatibility of the synthesized g-C$_3$N$_4$ QDs. Similar with the samples without A-CXL, seldom green spots can be observed in the images of the different groups with A-CXL, indicating that the 10 min's irradiation of 9 mW cm$^{-2}$ 365 nm UVA light on the cornea has little harm either the epithelium, the stroma, or the endothelium in vivo, providing significance information for the potential clinical applications.

## The in vivo safety of g-C₃N₄ QDs

Simultaneously, the in vivo safety of the proposed g-C$_3$N$_4$ QDs was also assessed. Initially, an assessment of the hemolysis potential of the g-C$_3$N$_4$ QDs dispersion was conducted, and the outcomes were displayed in Fig. 8a and Supplementary Fig. 21. The observations revealed that even at a concentration of 800 µg mL$^{-1}$, the hemolysis rate of red blood cells remained low, approximately 0.6%. This finding suggests that the obtained g-C$_3$N$_4$ QDs had minimal influence on immune regulation or heme response, thereby exhibited good blood biocompatibility.

Besides, blood samples were collected from rabbits at 1, 15, and 30 days post A-CXL treatment using g-C$_3$N$_4$ QDs dispersion. Routine blood analysis and biochemistry tests were conducted on these samples, with untreated rabbits serving as controls. The results of these tests are presented in Figs. 8b–d. In comparison to the control group, three indicators including ALT, ALB, PLT, exhibited abnormalities on both day 1 and day 15 following A-CXL treatment. Nevertheless, by the thirtieth day, these indicators had reverted to normal levels. It is noteworthy that all other blood parameters showed no statistically significant differences compared to the control group. Taken together, these findings collectively indicate that there was minimal to no significant inflammation or infection in rabbits treated with the synthesized g-C$_3$N$_4$ QDs as photosensitizers. The results strongly support the

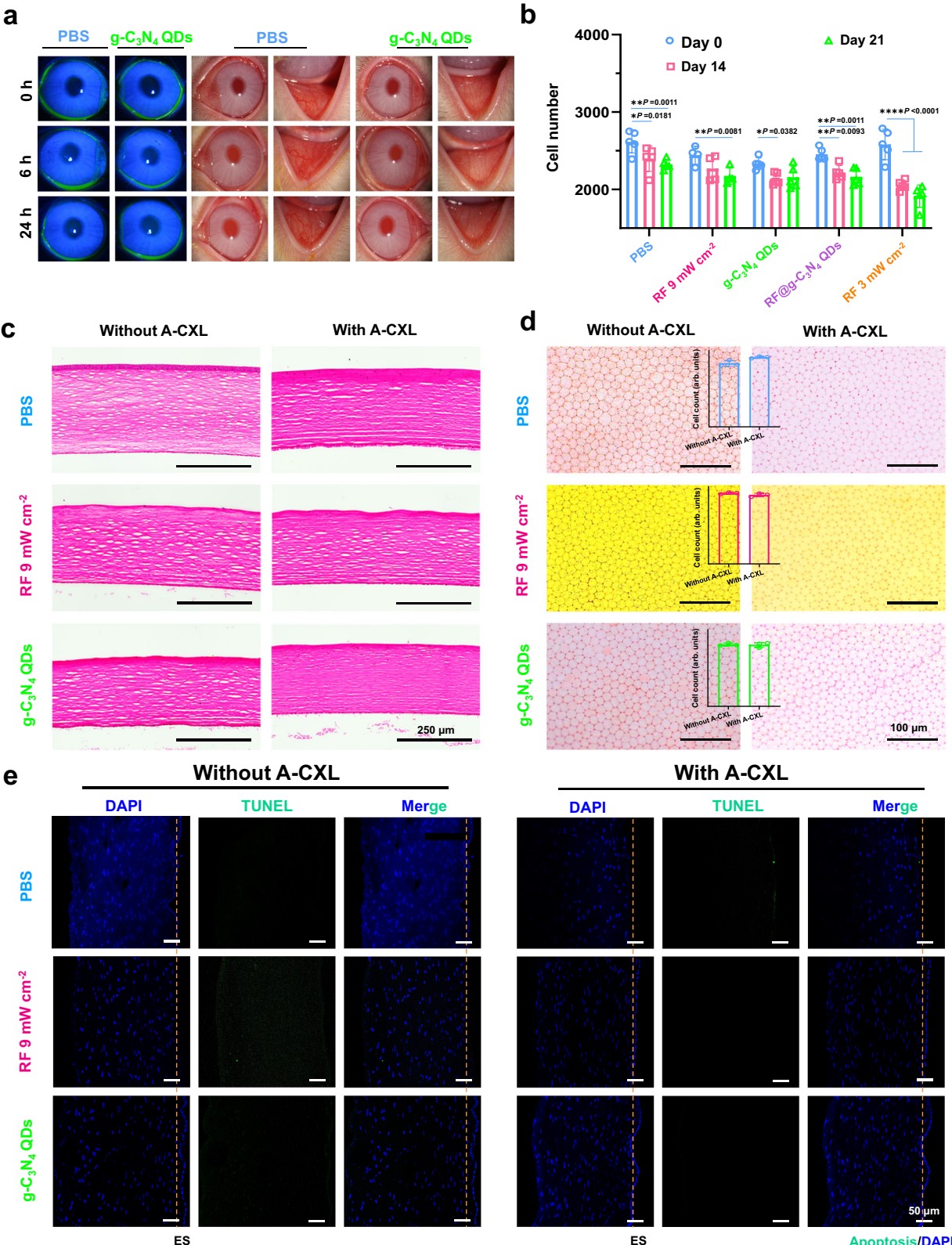

**Fig. 7 | The ocular biocompatibility of g-C₃N₄ QDs without or with A-CXL. a** Slit lamp images of the eyes treated with g-C₃N₄ QDs. **b** Corneal endothelial cell density on the 0, 1, 15, and 30 days (mean ± SD, $n = 4$, two-way ANOVA multiple comparison test, the absence of a *P*-value indicates no statistical difference between the groups ($p > 0.05$)). **c** Hematoxylin-Eosin staining (H&E) of the cornea ($n = 3$, similar results).

**d** Endothelial staining (ES) and the statistical analysis of ES photos (insert in Fig. 7b, mean ± SD, $n = 3$, the absence of a *P*-value indicates no statistical difference between the groups ($p > 0.05$)). **e** TUNEL test of cornea after being soaked with PBS, RF 9 mW cm⁻², and g-C₃N₄ QDs for 30 min ($n = 3$, similar results).

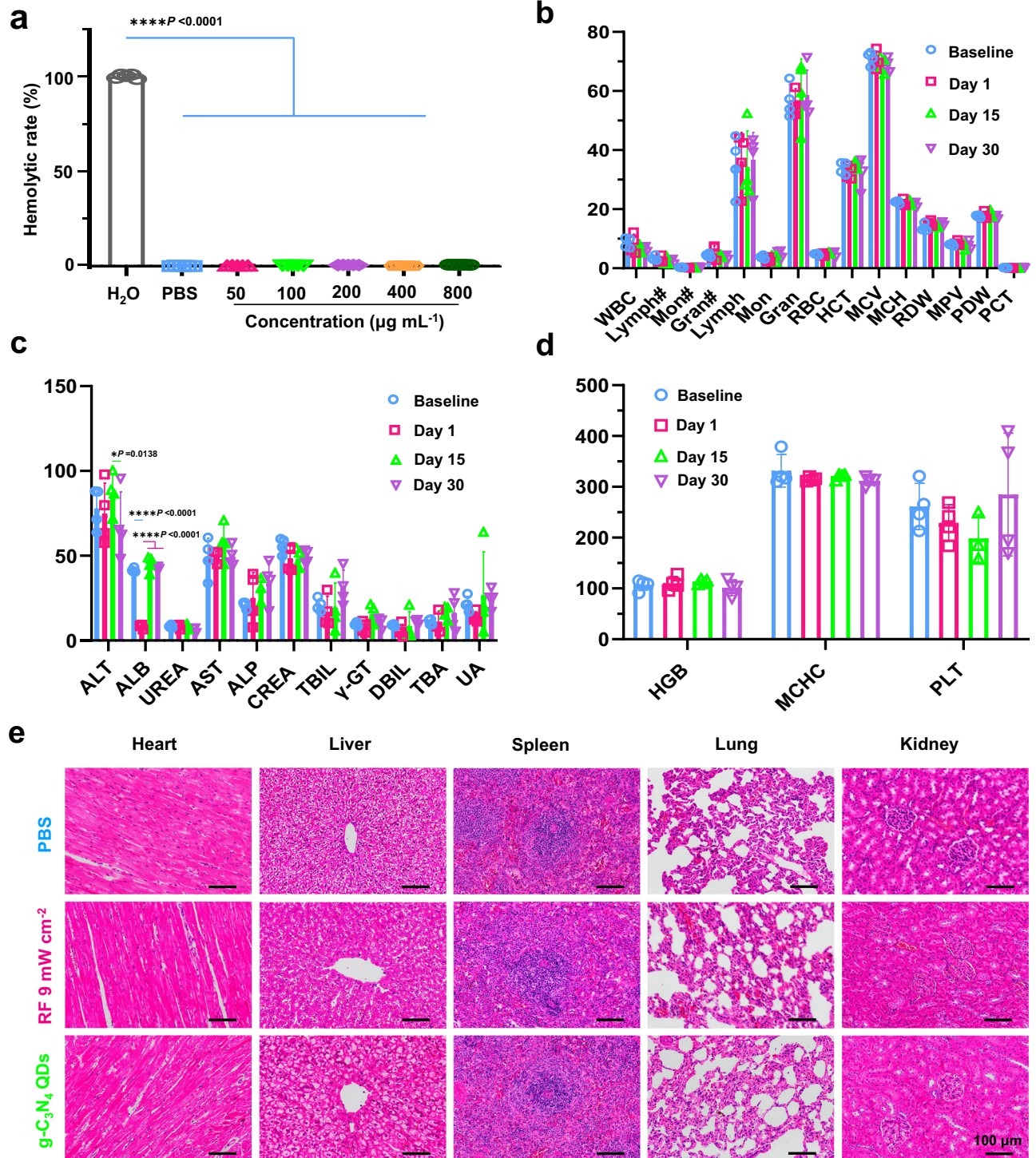

**Fig. 8 | The systemic biocompatibility of g-C₃N₄ QDs. a** Hemolysis experimental analysis of g-C₃N₄ QDs (mean ± SD, *n* = 5, two-way ANOVA multiple comparison test, the absence of a *P*-value indicates no statistical difference between the groups (*p* > 0.05)). **b–d** Routine blood analysis and biochemistry tests of the rabbits in g-C₃N₄ QDs group on days 0, 1, 15, and 30 days (WBC (10^9/L): White Blood cell count; Lymph# (10^9/L): Lymphocyte count; Mon# (10^9/L): monocyte count; Gran# (10^9/L): Granulocyte count; Lymph (%): Lymphocyte percentage; Mon (%): monocyte percentage; Gran (%): Granulocyte percentage; RBC(10^12/L): Red Blood cell count; HCT(%): Hematocrit; MCV(fL): Mean corpuscular volume; MCH (pg): Mean corpuscular hemoglobin; RDW (%): Red cell distribution width; MPV(fL): Mean platelet volume; PDW: Platelet distribution width; PCT: Plateletcrit;

ALT (U/L): Alanine transaminase; ALB (g/L): Albumin/Albumin blood test; UREA (mmol/L) Urea/urea test; AST (U/L): Aspartate aminotransferase; ALP (U/L): Alkaline phosphatase; CREA (mmol/L): Creatinine/creatinine test; TBIL (mmol/L): Total bilirubin; γ-GT (U/L): Gamma-glutamyl transferase; DBIL (mmol/L): Direct bilirubin; TBA Total bile acid/Total bile acid test, UA Uric acid/uric acid test, HGB (g/L) Hemoglobin, MCHC (g/L) Mean corpuscular hemoglobin concentration, PL (g/L)T Platelets count, *n* = 4, two-way ANOVA multiple comparison test, the absence of a *P*-value indicates no statistical difference between the groups (*p* > 0.05)). **e** Hematoxylin-Eosin staining (H&E) of the different organs of the rabbits in g-C₃N₄ QDs and RF 9 mW cm⁻² groups on days 30.

safety and potential clinical applicability of g-C$_3$N$_4$ QDs as promising A-CXL agents.

At the same time, multiple tissues, including the heart, liver, spleen, lung, and kidney, were collected at 30 days post A-CXL treatment using PBS, RF solution and g-C$_3$N$_4$ QDs dispersion. H&E staining were carried out, with untreated rabbits serving as controls. As depicted in Fig. 8e, the results demonstrated that there were no observable pathological alterations in the heart, liver, spleen, lung, or kidney over the course of treatment. Specifically, liver samples exhibited normal hepatocytes, lung samples showed no signs of pulmonary fibrosis, kidney sections revealed a clearly visible glomerulus structure, and there was no evidence of necrosis in any of the histological samples. These findings strongly suggest that g-C$_3$N$_4$ QDs have minimal side effects and possess great clinical applicability as A-CXL agents.

### The functional mechanism of the g-C$_3$N$_4$ QDs in A-CXL

In order to study the possible mechanism by which g-C$_3$N$_4$ QDs functions during the A-CXL process, the A-CXL effect of the g-C$_3$N$_4$ QDs photosensitizer in a hypoxic environment was investigated (A simple scheme of the operation was shown in Fig. 9a). A 9 mW cm$^{-2}$ UVA laser with 10 min irradiation time was used. Enzymatic digestion of the corneas was performed to simply evaluate the A-CXL effect under hypoxia conditions, and the results are shown in Figs. 9b–d. Compared with PBS, the application of RF as the photosensitizer resulted in a weak A-CXL effect under hypoxic conditions, it took only about 8 h for the treated cornea tissues to dissolve completely. These results further verified the importance of adequate oxygen in the CXL process using RF as the photosensitizer. When g-C$_3$N$_4$ QDs was used as the photosensitizer for CXL, after 48 h digestion, a residual corneal disc with an area about 46% of its initial size was left, even larger than that in a normal oxygen environment, indicating the excellent A-CXL effect using g-C$_3$N$_4$ QDs as a photosensitizer in a hypoxic environment. While comparing with g-C$_3$N$_4$ QDs group, the addition of RF to the g-C$_3$N$_4$ QDs aqueous dispersion has little effect on the further improvement of the A-CXL effect (Fig. 9c). The average dry weight of the residual cornea after 48 h digestion of different groups further confirms this conclusion (Fig. 9d). Meanwhile, comparing with the dry weight obtained in a normal oxygen environment for the g-C$_3$N$_4$ QDs group, the average dry weight of the residual cornea after 48 h digestion was much lower in a hypoxic environment, only about 50% of the dry weight in normal oxygen environment, indicating the hypoxia conditions can still influence the A-CXL effect, though the existence of g-C$_3$N$_4$ QDs relieved the degree of hypoxia.

Compared to the g-C$_3$N$_4$ QDs group, the little improvement of the A-CXL effect with the addition of RF to the g-C$_3$N$_4$ QDs aqueous dispersion may be due to inadequate oxygen supplementation using g-C$_3$N$_4$ QDs as photocatalytic oxygen generation agents, it is possible that they cannot supply enough O$_2$ for both RF and g-C$_3$N$_4$ QDs consumption. Besides, under the excitation of the same UVA intensity, the intensity variation of the excitation spectra of different groups further confirmed the internal energy competition between g-C$_3$N$_4$ QDs and RF. As displayed in Supplementary Fig. 22, for the same concentration of RF, monitored at 525 nm, it can be found that the intensity of the RF@g-C$_3$N$_4$ QDs excitation spectrum is lower than that of RF, combining with the UV−visible absorbance spectra of g-C$_3$N$_4$ QDs, the decrease of the RF excitation spectra in the RF@g-C$_3$N$_4$ QDs composite should be due to the energy absorbance of the g-C$_3$N$_4$ QDs. Vice versa, the existence of the RF also can influence the energy absorbance of g-C$_3$N$_4$ QDs as well as the subsequent O$_2$ and $^1$O$_2$ supply, resulting the similar A-CXL effect between g-C$_3$N$_4$ QDs and RF@g-C$_3$N$_4$ QDs groups, especially in a hypoxic environment.

Based on the aforementioned experimental results, it is easy to conclude that the g-C$_3$N$_4$ QDs play an important role during the A-CXL process, especially in a hypoxic environment. Combined with its photocatalytic oxygen generation property, a possible functional mechanism of g-C$_3$N$_4$ QDs used as photosensitizer in the A-CXL process is presented in Supplementary Fig. 23. In a hypoxic environment, O$_2$ is firstly generated by decomposition of H$_2$O using g-C$_3$N$_4$ QDs as a catalyst under the irradiation of UVA light. The produced O$_2$ molecules can be further reduced by the excited electrons under UVA irradiation to generate $^1$·O$_2^-$ in the confined space within the stroma and the existence of g-C$_3$N$_4$ QDs. Part of the produced $^1$·O$_2^-$ can interact with excited holes to give $^1$O$_2$, and further enhance the CXL effect. Thus, in order to full play the dual function of g-C$_3$N$_4$ QDs, proper UVA intensity and irradiation time are needed.

In this work, photosensitizers based on g-C$_3$N$_4$ QDs-based oxygen self-sufficient nanoplatform (g-C$_3$N$_4$ QDs or RF@g-C$_3$N$_4$ composite photosensitizers) were synthesized, with the purpose of solving the hypoxia problem during CXL. The obtained g-C$_3$N$_4$ QDs exhibited excellent photocatalytic oxygen generation ability, high ROS yield and excellent biosafety. The biomechanical stiffening and enhanced enzymatic resistance effects of the 9 mW cm$^{-2}$ (10 min) A-CXL procedure were enhanced obviously by replacing the riboflavin photosensitizer with g-C$_3$N$_4$ QDs or RF@g-C$_3$N$_4$ QDs composite, even comparable to the conventional RF CXL protocol (3 mW cm$^{-2}$, 30 min). The increased A-CXL effect was even more notable in a hypoxic environment, due to its excellent photocatalytic oxygen generation properties of g-C$_3$N$_4$ QDs, and this indicates a potential application in clinical settings. Meanwhile, the roles played by g-C$_3$N$_4$ QDs during the A-CXL process were also studied, which confirmed the need for the use of higher UVA intensities of 6 mW cm$^{-2}$ and above to allow the photosensitizer to achieve its dual functions. Thus, the g-C$_3$N$_4$ QDs may potentially be applied in A-CXL for corneal ectasias.

## Methods

The animal study was approved by the Ethical Committee of EYE and ENT Hospital of Fudan University (IACUC-DWZX-2023-026), and conducted in strict adherence to the ARVO Statement for the Use of Animals in Ophthalmic and Vision Research.

### Materials and reagents

Melamine, KOH, NaOH, potassium bromide (KBr), ethanol, trichloroacetic acid, paraffin, and magnesium acetate tetrahydrate were bought from Aladdin Industrial Inc.in Shanghai, China. Trypan Blue, Riboflavin 5′-phosphate sodium (RF), and Alizarin Red S were acquired from Sigma Aldrich in MI, Italy. The Cell Counting Kit-8, Calcein-AM, PI, Positive Control Preparation Kit for TUNEL Assay and In Situ Cell Death Detection Kit Fluorescein were purchased from Beyotime Biotechnology Inc.in Shanghai, China. The Annexin V-FITC/PI Apoptosis Detection Kit was sourced from Vazyme Biotech Co., Ltd. in Nanjing, China. Collagenase Type 2 was bought from Worthington Biochemical Corporation in the United States. All reagents utilized in this study were used directly without undergoing further purification.

### Synthesis of g-C$_3$N$_4$ QDs and RF@g-C$_3$N$_4$ QDs composite photosensitizers

**Synthesis of g-C$_3$N$_4$ QDs photosensitizers.** g-C$_3$N$_4$ QDs photosensitizer was synthesized using a modified method reported by Krivtsov et al. [35]. In a typical process: 3 g of melamine was fully mixed with 20 mmol of KOH and 10 mmol of NaOH in 2 mL of anhydrous ethanol, followed by grinding for 20 min to get a homogenous white powder. This powder was then transferred into a crucible and annealed at different temperature (330−370 °C) in a tubular furnace for 2 h in air. The rate of temperature increase was controlled at 2 °C min$^{-1}$ to 100 °C, and 5 °C min$^{-1}$ from 100 °C to the final desired temperature. After naturally cooling to room temperature, the obtained yellow powder was dispersed in deionized water, large particles were removed by filtering using a 0.2 μm filter membrane. This yielded g-C$_3$N$_4$ QDs were further purified by dialyzing against 3 L of deionized water for a

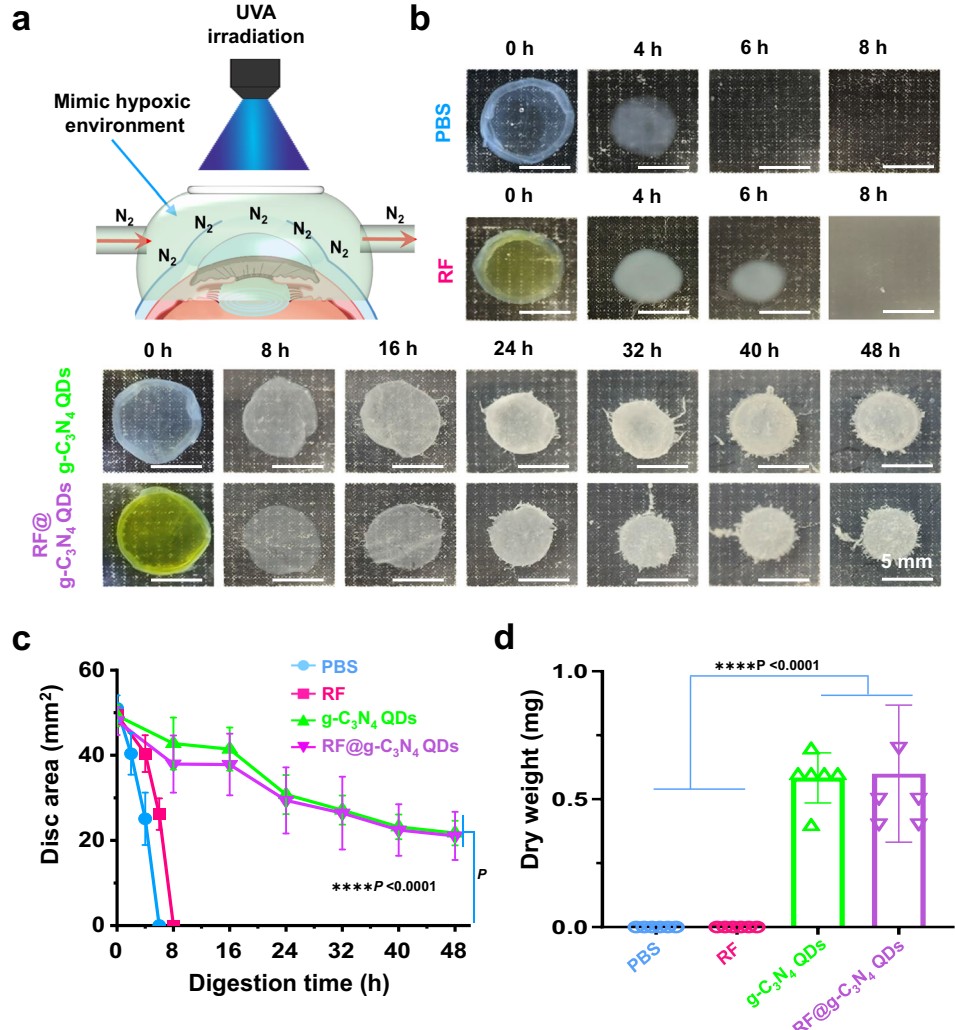

**Fig. 9 | The A-CXL efficacy under hypoxic environment. a** Establishment of hypoxic environment by pumping the $N_2$. **b** Photos of corneas after digestion in collagenase II for different times. **c** Measurements of the residual disc area over time (mean ± SD, $n = 6$, two-way ANOVA multiple comparison test, the absence of a *P*-value signifies no statistical significant difference between the groups ($p > 0.05$)). **d** The dry weight of the final residual cornea (mean ± SD, $n = 6$, two-way ANOVA multiple comparison test, the absence of a *P*-value indicates no statistical difference between the groups ($p > 0.05$)).

duration of 5 days, with the water being changed 3 times per day, and then either stored at 4 °C or freeze-dried in Labconco™ FreeZone™ 2.5 L machine.

### Synthesis of RF@g-$C_3N_4$ QDs composite photosensitizers
For a typical synthesis process of RF@g-$C_3N_4$ QDs composite photosensitizers, 2.5 mg freeze-dried g-$C_3N_4$ QDs and 0.1 mg Riboflavin-5-phosphate (RF) were dispersed in 1 mL of deionized water, which was stirred at room temperature for 1 h and subsequently stored at 4 °C for subsequent use.

### Synthesis of g-$C_3N_4$ bulk materials
For a typical synthesis process of g-$C_3N_4$ bulk materials, 5 g of melamine was added into a crucible and annealed in a tubular furnace at 550 °C for 2 h. The rate of temperature increase was controlled at 2 °C $min^{-1}$. Upon naturally cooling to room temperature, a yellow powder was obtained.

### Characterization
**Photocatalytic oxygen generation of g-$C_3N_4$ QDs.** A dissolved oxygen analyzer (DOS-1703, Shanghai Boqu Instrument Company Ltd, China) was used to detect the amounts of oxygen produced by g-$C_3N_4$ QDs under the irradiation of UVA light. The concentration of g-$C_3N_4$

QDs synthesized at different temperatures was controlled at 2.5 mg $mL^{-1}$ by dispersing the weighed 25 mg freeze-dried g-$C_3N_4$ QDs into 10 ml deionized water. Then, 6 mL aqueous dispersion of g-$C_3N_4$ QDs were firstly added into a 25 ml beaker with 5 mL liquid paraffin to prevent $O_2$ from the air entering the solution. After recording the initial oxygen content in the solution, the solution was then irradiated with a 15-minute exposure to 365 nm UVA light (16 W, ZF-20D Crypto UV analyzer, Shanghai Guanghao analytic instrument Co., Ltd). Following irradiation, the oxygen content of the dispersion was recorded again. The difference in the oxygen content values before and after irradiation corresponds to the amount of oxygen produced by g-$C_3N_4$ QDs under the irradiation of UVA. The process was repeated three times to obtain an average measurement of oxygen production.

Meanwhile, MC-SPB10 water-splitting reaction system (Merry Change Co., China) was further used to evaluate the photocatalytic $O_2$ evolution performance of g-$C_3N_4$ QDs obtained at 350 °C under 365 nm UVA irradiation with an intensity of 180 mW $cm^{-2}$ for a duration of 3 h using 10 vol% triethanolamine as a sacrificial agent.

### Singlet oxygen generation
To confirm the generation of singlet oxygen, a chemical probe, DPBF, was utilized. The absorption intensity at 410 nm was monitored via

UV–visible spectroscopy. In a typical DPBF experiment, a mixture of 180 µL aqueous solution of DPBF (100 µg mL$^{-1}$) and 20 µL g-C$_3$N$_4$ QDs aqueous solution was prepared. This mixture was kept in darkness for 1 h before being irradiated with a 9 mW cm$^{-2}$, 365 nm UV–visible light source for 10 min. During the irradiation process, the absorption intensity of DPBF at 410 nm was documented every 2 min. For comparative purposes, control groups were established: one group examined the absorption spectra of the same solution without UV–visible light irradiation, while another group examined the absorption spectra of a solution containing 180 µL DPBF aqueous solution and 20 µL H$_2$O without UV–visible light irradiation.

## In vitro biological compatibility evaluation

**Cytotoxicity evaluation.** The human retinal microvascular endothelial cells (HRMEC), human retinal epithelial cells (ARPE-19), human corneal epithelial cells (HCEC) were provided by the Global Bioresource Center. In vitro cytotoxicity measurements were conducted as follows: Initially, $1 \times 10^4$/well HCEC cells were seeded into a 96-well plate and incubated at 37 °C for 24 h to ensure cell adherence. Then, g-C$_3$N$_4$ QDs with different concentrations were introduced into the designated wells, with over six samples prepared and measured for each concentration group. The cells were then incubated for an additional 24 h at 37 °C, and the cell viability was assessed using a Cell Counting Kit-8 (CCK-8, CK-04, Dojindo). The same procedures were carried out for ARPE-19 and HRMEC cells.

The relative cell viability was calculated from optical density (OD) measurements according to the following equation:

$$\text{Cell viability}(\%) = \left( \text{OD}_{450,\text{sample}} - \text{OD}_{450,\text{blank}} \right) / \left( \text{OD}_{450,\text{control}} - \text{OD}_{450,\text{blank}} \right) *100\%$$

**Calcein-AM/PI double staining.** To investigate the impact of g-C$_3$N$_4$ QDs on HCEC cells, the Calcein-AM/PI double staining methods were further employed to discern the number of live and dead cells. Under a fluorescence microscope, living cells emit green fluorescence, whereas dead cells emit red fluorescence. After being co-cultured for 24 h with g-C$_3$N$_4$ QDs at different concentrations, Calcein-AM/PI solution was introduced into the culture medium. After an additional 15 min incubation, the distribution of the living and dead cells was observed and photographed using a Leica DMI8 inverted fluorescence microscope, and the experiments were repeated three times.

**Cell apoptosis assessment.** HCEC cells were firstly seeded onto 96-well plates and incubated for 24 h at 37 °C. Subsequently, the medium was then replaced with fresh medium containing g-C$_3$N$_4$ QDs with different concentrations. Following another 24 h of incubation period, the cells were harvested via trypsin digestion, washed with PBS and resuspended in 300 µL 1X binding buffer. Next, 4 µL of Annexin V-FITC and 1.5 µL of PI were added in sequence. After being kept in dark for another 15 min, all samples were analyzed using an Accuri C6 flow cytometer to evaluate cell viability, and the experiments were repeated three times.

## In vivo evaluation

Male New Zealand white rabbits (weight, 2.0–2.5 kg, 4 months) were raised with unrestricted access to food and water for the entire duration of the test period.

## In vivo CXL evaluation

Initially, the integrity of the corneal epithelium was examined using slit lamp biomicroscopy (Kanghua, China). Subsequently, rabbits were anesthetized with xylazine hydrochloride (0.2 mL kg$^{-1}$) and 2% pentobarbital sodium (1 mL kg$^{-1}$) administered via intramuscular injection. Following anesthesia, a blunt knife was utilized to remove the corneal epithelium, after which the cornea was presoaked for 30 min with one of the following solutions according to its treatment group: PBS (group was presoaked with PBS solution), RF 9 mW cm$^{-2}$ group was presoaked with 0.1% RF in PBS solution, g-C$_3$N$_4$ QDs group was presoaked with 2.5 mg mL$^{-1}$ g-C$_3$N$_4$ QDs aqueous dispersion, RF@g-C$_3$N$_4$ QDs group was presoaked with 2.5 mg mL$^{-1}$ g-C$_3$N$_4$ QDs and 0.1% RF aqueous dispersion, while RF 3 mW cm$^{-2}$ group was presoaked with 0.1% RF in 20% Dextran T500 solution.

After presoaking for 30 min, the cornea was irradiated using 365 nm continuous UVA light with an 8 mm diameter facula. The UVA intensity and irradiation time varied among groups. For the g-C$_3$N$_4$ QDs group, the following irradiation protocols were used: 3 mW cm$^{-2}$ (30 min), 6 mW cm$^{-2}$ (15 min), 6 mW cm$^{-2}$ (30 min), 9 mW cm$^{-2}$ (10 min), 9 mW cm$^{-2}$ (15 min), 9 mW cm$^{-2}$ (20 min) and 18 mW cm$^{-2}$ (5 min) UVA. The PBS, RF 9 mW cm$^{-2}$ and RF@g-C$_3$N$_4$ QDs groups underwent irradiation with 9 mW cm$^{-2}$ (10 min) UVA. 3 mW cm$^{-2}$ (30 min) UV.

**Enzymatic digestion of collagen.** An 8 mm disk was trephined from the center of corneas in each of the irradiated and non-irradiated treatment groups described above. These disks were then immersed in 3 mL 0.2% Collagenase Type 2 solution and agitated continuously (175 rpm, 37 °C). Throughout the digestion process, images of the corneal tissue were captured every 2 h until complete digestion of the PBS group occurred. After this point, images of the corneal tissue in the remaining experimental groups and RF group were taken every 8 h over a total period of 48 h. These images were later analyzed using Image J to determine the remaining corneal tissue area for each group at various time point[37]. After 48 h of digestion, all remaining corneal tissue was dried in an oven at 60 °C for 3 d to measure the undigested dry tissue mass for each treatment group ($n \geq 3$ per group).

**Stress–strain measurements.** After euthanasia, the corneal thickness measurements were initially taken using the SP-3000 A ultrasonic biometers for ophthalmolog scanner (Tomox, Japan). Subsequently, A tailor-made double-bladed cutter was utilized to cut a 10 mm × 3 mm strip along the superior-inferior meridian from the center of non-irradiated PBS treated corneas, as well as UVA irradiated RF, g-C$_3$N$_4$ QDs and RF@g-C$_3$N$_4$ QDs treated corneas ($n \geq 3$ per group). Each strip was positioned vertically within the pneumatic jaws of the stretching equipment (Instron 68TM-30, Instron INC., MA, USA). Before conducting stress-strain measurements, the tissue underwent a preconditioning process. For pre-cycling, the tensile load was set at 0.5 N, with the tensile displacement increased to 1 mm at a rate of 2 mm min$^{-1}$, followed by a reduction to 0 N at the same rate. In total, three pre-cycles were conducted, with a 30-second interval between each. Immediately following preconditioning, each cornea strip was stretched to a 20% deformation at a consistent loading rate of 2 mm min$^{-1}$, and the corresponding stress–strain curve was documented. Finally, the Young's modulus was determined from the linear segment of each stress-strain curve.

**In vivo biomechanical evaluation.** As described previously, the A-CXL protocol was performed on the RF 9 mW cm$^{-2}$, g-C$_3$N$_4$ QDs and RF@g-C$_3$N$_4$ QDs groups, whereas conventional CXL protocol was executed on the RF 3 mW cm$^{-2}$ group ($n = 5$ per group). Then, corneal biomechanical parameters of the five groups were collected using the Corvis® ST, an advanced OCULUS corneal biomechanical analysis system. Data about pre-CXL (D0), D3, D5, D7, D14, and D21 time points were collected and analyzed. During data acquisition period, specific ophthalmic medications were administered. These included sodium hyaluronate eye drops (HYLO-COMOD®), dexamethasone tobramycin eye ointment (Tobra-Dex®), and deproteinized calf blood extract eye gel.

**CCT measurements.** During the collection of the data about in vivo biomechanical parameters, the central corneal thickness (CCT) of the corneas was also measured in the five groups following CXL at various intervals (21, 30 days) using the Optovue RTVue OCT ($n = 4$).

## Assessment of in vivo biocompatibility

The biocompatibility of g-C$_3$N$_4$ QDs in vivo was evaluated both in short-term and long-term through a comprehensive suite of tests. These included slit lamp biomicroscopy for detailed ocular examination, specular microscope examination to assess endothelial cell health, corneal hematoxylin and eosin (H&E) for histological analysis, Alizarin Red S and Trypan Blue endothelial staining (ES) to visualize the endothelium, corneal TUNEL staining to detect apoptotic cells. Additionally, routine blood analysis, biochemical tests, and H&E staining of various organs at designated intervals following A-CXL treatment were conducted, besides the hemolysis experiment of g-C$_3$N$_4$ QDs.

**Ocular acute irritation response of g-C$_3$N$_4$ QDs.** The ocular acute irritation response of g-C$_3$N$_4$ QDs was examined in rabbits via the modified Draize test. For this test, 50 µL of g-C$_3$N$_4$ QDs (2.5 mg mL$^{-1}$) was instilled into the lower conjunctival sac of the rabbits' right eyes, while their left eyes were treated normal saline as a control. Ocular manifestations were recorded using a slit lamp microscope at various intervals: baseline (0 h), immediately after instillation (0 h), 6 h post-instillation, and 24 h post-instillation ($n = 3$). Additionally, fluorescein staining was employed to observe the integrity of the corneal epithelial.

**Postoperative corneal recovery evaluation.** Following A-CXL treatment using g-C$_3$N$_4$ QDs dispersion, corneal recovery was monitored using a slit-lamp microscope, with the RF 9 mW cm$^{-2}$ group served as a control. Photographic documentation was collected and compared at intervals of 0, 1, 3, 7, 14, and 30 days ($n \geq 3$ per group).

**Endotheloscopy.** During the acquisition of the data about in vivo biomechanical parameters, endothelial cell density of the cornea was recorded using specular microscope (EM-4000) on days 0, 1, 15, and 30. The data obtained was then subjected to comparative analysis across various groups ($n = 4$ per group).

**Alizarin Red S and Trypan Blue staining of the endothelium.** A 5 mm full-tissue thickness biopsy was obtained from the center of PBS-treated corneas, as well as RF, g-C$_3$N$_4$ QDs, and RF@g-C$_3$N$_4$ QDs corneas, without and with exposure to 9 mW cm$^{-2}$ (10 min) UVA irradiation ($n = 3$ per group). The biopsy was then placed on a glass slide with the endothelium side up. Subsequently, 0.2% trypan blue was applied to the endothelium for 90 s, followed by 2–3 washes with PBS and gently draining to eliminate any excess solution. Next, alizarin red S (0.2%, pH 4.7) was added to the endothelium for an additional 90 s, after which the sample was rinsed 2–3 times with PBS. Finally, the endothelium morphology was examined using a light microscope (Leica DM750).

**Hematoxylin-eosin staining of cornea.** Immediately following euthanasia, eyes treated with both nonirradiated and 9 mW cm$^{-2}$ (10 min) UVA irradiated g-C$_3$N$_4$ QDs ($n = 3$ per group) were transferred into a fixative solution for 2 h. This fixative consisted of 80% ethanol, 10% acetic acid and 10% formaldehyde. Subsequent to a dehydration process, a 5 µm full-tissue thickness section was obtained from the cornea's center. The tissue section was then covered with hematoxylin for 10 min, rinsed with PBS, and stained with eosin for 8 min to visualize the extracellular matrix. The corneal epithelium and stromal cells were then examined in the light microscope (Leica DM750).

**TUNEL staining of cornea.** Apoptosis assay was done by TUNEL staining. A 10 µm-thick frozen tissue section was taken from the center of non-irradiated ($n = 3$) and 9 mW cm$^{-2}$ (10 min) UVA irradiated ($n = 3$) g-C$_3$N$_4$ QDs treated corneas and soaked with 4% paraformaldehyde for 15 min after which, the sample was rinsed 2–3 times using PBS. Then, 0.1% TritonX was added dropwise to cover the tissue section for 5 min, and the section subsequently washed using PBS for 2–3 times. The tissue section was further soaked in diluted reaction buffer (180 µL PBS

and 20 µL reaction buffer) for 15 min, after which 10 µL dUTP (LS) solution was added. The tissue section maintained in dark wet box at 37 °C for 1 h, and rinsed 5 times using PBS. Finally, 15 µL DAPI solution was added dropwise to cover the tissue section, the obtained tissue section was further sealed and stored at 4 °C avoid light. The TUNEL-positive control was also performed similarly as above, except 10 uL DNA enzyme solution (mixture of 1 µL DNaseI, 90 µL PBS and 10 µL reaction buffer) was added after 5 min soak in diluted reaction buffer.

**Hemolysis experiment.** To test the blood biocompatibility of g-C$_3$N$_4$ QDs, rabbit blood was firstly collected and subjected to centrifugation at 2000 g/min at 4 °C for 10 min to separate the red blood cells, which was further collected and resuspended in cold PBS solution undergoing a second centrifugation at the same conditions. The obtained red blood cells were further washed and diluted with PBS solution by 10 times. Then, 0.1 mL of the diluted red blood cells were added to 0.4 mL of PBS solution (negative control group), deionized water (positive control group), and g-C$_3$N$_4$ QDs dispersion with different concentrations (sample group). The samples were incubated in a controlled environment at 37 °C for 4 h, followed by centrifugation at 300 g/min at 4 °C for 10 min. Finally, the supernatant's absorbance was precisely measured at a wavelength of 541 nm using an Agilent Synergy H1 microplate reader ($n = 5$ per group). The hemolysis percentage was then calculated using the following formula:

$$\text{Hemolysis}(\%) = \left( A_{\text{sample}} - A_{\text{negative control}} \right) / \left( A_{\text{positive control}} - A_{\text{negative control}} \right).$$

**Routine blood analysis and biochemistry tests.** During the collection of the data about in vivo biomechanical parameters, blood samples were also collected carefully from the central artery of each rabbit's ear using a 5-gauge needle. This procedure was repeated on days 0, 1, 15, and 30 days, with 3–5 ml of blood being collected each time. EDTA vacuum tubes were used for collection, and the blood was continuously agitated to prevent coagulation. The blood samples were then stored at 4 °C for further processing. Meanwhile, 1.5–2 ml of the serum was separated from the whole blood by centrifuging at 3000 rpm for 15 min and stored frozen at −20 °C for subsequent analysis. The blood samples were utilized for routine blood tests using Rayto Chemray 420, providing valuable insights into the rabbits' hematological status. While the serum samples were specifically earmarked using Mindray BC-2800vet for liver and kidney biochemistry function tests, enabling a comprehensive assessment of the rabbits' physiological responses ($n = 4$ per group).

**Haematoxylin and eosin stain (H&E stain).** During the treatment process, various organs were also collected for histopathological examination. Following a 10% formalin fixation, the organs were dehydrated in a graded series of ethanol solutions, ensuring the complete removal of moisture. The dehydrated organs were then embedded in paraffin, which served as a stable matrix for sectioning. The paraffin-embedded organs were then precision-cut into 4–5 µm sections. Following a series of dehydration sections and other preparations, hematoxylin was applied dropwise to the sections for 10 min before being rinsed thoroughly with PBS to remove any excess stain. Subsequently, eosin was employed to specifically stain the extracellular matrix for 8 min, ensuring optimal staining intensity. The resulting stained sections were then carefully examined under a high-resolution light microscope (Leica DM750), revealing intricate details of the organ's histopathology ($n = 3$ per group).

## Other characterizations

The phase structure and crystallographic state of the obtained g-C$_3$N$_4$ QDs under various conditions were examined using a Smartlab 9 powder Diffractometer with Cu K radiation (λ = 1.54 Å) manufactured by Hitachi, Japan. The morphology and microstructure of these QDs were inspected via a high-resolution transmission electron microscope

(HRTEM, JEM2100F, Japan) operating at an accelerating voltage of 200 kV. The surface functional groups present on the products were analyzed by Fourier-transform infrared (FTIR) spectroscopy, covering a range of 400 to 4000 cm$^{-1}$. These spectra were obtained using a Thermal Fisher Nicolet 6700 spectrometer (USA) using the KBr pellet technique. The UV–visible absorbance spectra of the sample were determined using UV–visible spectrophotometry (UV–Vis, Agilent Cary100). Additionally, excitation and emission spectra were captured using a Hitachi F-7000 Fluorescence Spectrophotometer (Japan). The Zeta potential and size distribution of the g-C$_3$N$_4$ QDs were measured using Malvern Zetasizer equipment (Nano-ZS90, UK). Furthermore, X-ray photoelectron spectra experiments were conducted on a Thermo Scientific Escalab 250 Xi spectrometer, employing monochromatic Al Kα radiation (hv = 1486.6 eV) as the excitation source. During XPS measurements, the test spot area was set to 500 μm, with the tube current maintained at 10 mA. Photoelectrons were selected in energy using a hemispheric electron analyzer, and all binding energy (BE) values were referenced to the C 1 s peak of carbon at 284.6 eV.

## Statistical analysis

All data were presented as the mean ± standard deviation (means ± SD) from replicated experiments or samples within a representative experiment. Statistical significance between groups was analyzed using Graphpad Prism 8.0 software, employing unpair $t$-test, one-way analysis of variance (ANOVA) or two-way ANOVA. Detailed information regarding sample size, comparisons methods, and other specifics are provided in the legend of each figure.

## Reporting summary

Further information on research design is available in the Nature Portfolio Reporting Summary linked to this article.

## Data availability

The authors declare that all data supporting the findings of this study are accessible within the article and its Supplementary Information. Source data are provided with this paper.

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

## Acknowledgements
This work was supported in part by the Project of National Natural Science Foundation of China (No. 82271048, J Huang; 82301215, G Zhao), Shanghai Science and Technology (No. 23ZR1409500, M Yang; 23S11900200, M Yang; 23XD1420500, J Huang; 22S11900200, J Huang), EYE & ENT Hospital of Fudan University High-level Talents Program (No. 2021318, J Huang), Clinical Research Plan of SHDC (No. SHDC2020CR1043B, X Zhou), Medical Engineering fund of Fudan University (No. yg2023-06, J Huang; yg2023-26, X Zhou), Project of Shanghai Xuhui District Science and Technology (No. 2020-015, X Zhou), Shanghai Anticancer Association EYAS PROJECT (SACA-CY22C03, Z Chen), Program for Professor of Special Appointment (Eastern Scholar, No. TP2022046, J Huang) at Shanghai Institutions of Higher Learning. K M Meek and S Hayes were supported by the Medical Research Council (UK) grant number MR/S037829/1. The funders played no part in the study design, data collection and analysis, decision to publish, or preparation of the manuscript.

## Author contributions
J Huang, M Yang and X Zhou conceptually designed the studies. M Yang, J Huang, G Zhao interpreted the results. M Yang, T Chen, X Chen, H Pan and N Zhao performed the in vitro materials synthesis and characterization, X Ma, and Y Wu provided assistance, M Yang analyses materials characterization results. G Zhao, X Chen, Z Chen and N Zhao, performed in vitro cell biocompatibility evaluation, T Chen, X Chen, H Pan, S Zhang, Q Ye, M Chen and X Li performed animal CXL surgeries and collected tissue samples. M Yang, J Huang, G Zhao, Z Chen and R Gao analysed the in vivo evaluation results. M Yang authored the manuscript draft, incorporating contributions from all authors. M Yang, Z Chen, Y Zhang, N Kong, J Huang and W Tao edited the manuscript. K M Meek and S Hayes provided consultation, data evaluation and polish the manuscript. All authors commented on and revised the paper.

## Competing interests
The authors declare no competing interest.

## Additional information

[1]Eye Institute and Department of Ophthalmology, Eye & ENT Hospital, Fudan University, NHC Key Laboratory of Myopia and Related Eye Diseases; Key Laboratory of Myopia and Related Eye Diseases, Chinese Academy of Medical Sciences; Shanghai Research Center of Ophthalmology and Optometry, Shanghai 200030, China. [2]School of Ophthalmology and Optometry and Eye Hospital, Wenzhou Medical University, Wenzhou, Zhejiang 325027, China. [3]School of Chemical Engineering, Northeast Electric Power University, Jilin 132000, China. [4]School of Optometry and Vision Sciences, Cardiff University; Cardiff Institute for Tissue Engineering and Repair School of Pharmacy and Pharmaceutical Sciences, Cardiff University, Redwood Building, King Edward VII Avenue, Cardiff CF10 3NB, UK. [5]School of Environmental Science and Engineering, Nanjing University of Information Science and Technology, Nanjing 210044, China. [6]Center for Nanomedicine and Department of Anesthesiology, Brigham and Women's Hospital, Harvard Medical School, Boston, MA 02115, USA. [7]These authors contributed equally: Mei Yang, Tingting Chen, Xin Chen, Hongxian Pan, Guoli Zhao. ✉e-mail: meiyang@fudan.edu.cn; xingtaozhou@fudan.edu.cn; jinhaihuang@fudan.edu.cn

