## [Peer Review File · Nature Communications]

Reviewers' Comments:

Reviewer #1:

Remarks to the Author:

The manuscript by Yang et al highlights a very interesting biomedical approach of a nano-photocatalyst- g-C₃N₄ QDs. Keratoconus is a challenging clinical condition and proposing g-C₃N₄ QDs for corneal crosslinking is a novel approach for the field.

When a nanomaterial is tested for a biomedical application, it would be really important to assess the stability of the material. What is the stability profile of g-C₃N₄ QDs over time? How does biological environment/fluid change the stability of this material, considering its suggested use for cornea? Does the nanomaterial aggregate or precipitate over time before application?

For cornea, transparency is an essential parameter. On the other hand, full dispersibility of nanomaterial is a limiting factor. What is the effect of the g-C₃N₄ QDs on the corneal transparency?

More emphasis can be made for material choice. Why did the authors select this nano photocatalyst out of others available in literature? Why did they select the quantum dot version instead of the nanosheet format? How is it compared with other nano based corneal crosslinking studies?

In the results section, it might be beneficial to delve into the clinical relevance of your study's findings. Specifically, consider explaining how the observed biocompatibility of g-C₃N₄ QDs within the cornea is directly linked to potential applications in clinical practice, particularly in the context of corneal cross-linking (A-CXL). How will this application contribute to the future of the cornea department, and does it have the potential to enable office-based corneal cross-linking (CXL)? How is g-C₃N₄ QD based crosslinking compared to clinically applied strategies?

It appears that there is a typographical error in Figure 6's footnote. The correct term should be "Tukey" multiple comparisons test, not "Turkey." Here's the corrected footnote:

Figure 6 footnote: "Statistical analysis was performed using one-way ANOVA with Tukey multiple comparisons test"

Reviewer #2:

Remarks to the Author:

This work is novel and and may have a significant positive impact on the field of cornea and collagen crosslinking. It is previously reported that additional supply of ex-situ oxygen to the cornea (e.g., outside the cornea via goggles, facemask) increases the crosslinking yield. What is noteworthy about this work is that the oxygen supply is done in-situ (e.g. inside the cornea) enhancing availability of oxygen at the crosslinking site.

Despite the fact that the concept is novel, the experimental work do not fully support the conclusions and additional evidence is needed, for instance:

1. the HCEC cells shown in Fig 3 (a) (in vitro cell viability study) seem round and not elongated suggesting poor cell biocompatibility. Also, by increasing the g-C₃N₄ QDs concentration, the number of live cells seem to decrease while not obvious in Fig.3 (e). This may be due to poor quality of the images.
2. the authors have used their g-C₃N₄ QDs in vivo in rabbits, however the degradation and mechanical tests are done on ex-vivo samples and no in vivo testing such as biomechanics (e.g., Corvis® ST) OCT for thickness measurement, or in vivo toxicity tests are not performed. Given the nanosized nature of the g-C₃N₄ QDs, further in vivo safety and efficacy testing are required to ensure they pose no risk to human. The regulatory requirements for nanoparticle based medical products are highly stringent as the nanoparticles can travel in the body or enter the cells and their impacts must be evaluated.

Overall, I suggest that further in vitro and in vivo testing (as mentioned above) are required to make the manuscript publishable in Nature Communications.

Development of graphitic carbon nitride quantum dot-based platform for enhanced corneal crosslinking

REVIEWER COMMENTS

Reviewer #1 (Remarks to the Author):

The manuscript by Yang et al highlights a very interesting biomedical approach of a nano-photocatalyst- g-C₃N₄ QDs. Keratoconus is a challenging clinical condition and proposing g-C₃N₄ QDs for corneal crosslinking is a novel approach for the field. When a nanomaterial is tested for a biomedical application, it would be really important to assess the stability of the material. What is the stability profile of g-C₃N₄ QDs over time? How does biological environment/fluid change the stability of this material, considering its suggested use for cornea? Does the nanomaterial aggregate or precipitate over time before application? For cornea, transparency is an essential parameter. On the other hand, full dispersibility of nanomaterial is a limiting factor. What is the effect of the g-C₃N₄ QDs on the corneal transparency? More emphasis can be made for material choice. Why did the authors select this nano photocatalyst out of others available in literature? Why did they select the quantum dot version instead of the nanosheet format? How is it compared with other nano based corneal crosslinking studies? In the results section, it might be beneficial to delve into the clinical relevance of your study's findings. Specifically, consider explaining how the observed biocompatibility of g-C₃N₄ QDs within the cornea is directly linked to potential applications in clinical practice, particularly in the context of corneal cross-linking (A-CXL). How will this application contribute to the future of the cornea department, and does it have the potential to enable office-based corneal cross-linking (CXL)? How is g-C₃N₄ QD based crosslinking compared to clinically applied strategies? It appears that there is a typographical error in Figure 6's footnote. The correct term should be "Tukey" multiple comparisons test, not "Turkey." Here's the corrected footnote: Figure 6 footnote: "Statistical analysis was performed using one-way ANOVA with Tukey multiple comparisons test"

Reviewer #2 (Remarks to the Author):

This work is novel and may have a significant positive impact on the field of cornea and collagen crosslinking. It is previously reported that additional supply of ex-situ oxygen to the cornea (e.g., outside the cornea via goggles, facemask) increases the crosslinking yield. What is noteworthy about this work is that the oxygen supply is done in-situ (e.g. inside the cornea) enhancing availability of oxygen at the crosslinking site.

Despite the fact that the concept is novel, the experimental work do not fully support the conclusions and additional evidence is needed, for instance:

1. the HCEC cells shown in Fig 3 (a) (in vitro cell viability study) seem round and not elongated suggesting poor cell biocompatibility. Also, by increasing the g-C₃N₄ QDs concentration, the number of live cells seem to decrease while not obvious in Fig.3 (e). This may be due to poor quality of the images.

2. the authors have used their g-C₃N₄ QDs in vivo in rabbits, however the degradation and mechanical tests are done on ex-vivo samples and no in vivo testing such as biomechanics (e.g., Corvis® ST) OCT for thickness measurement, or in vivo toxicity tests are not performed. Given the nanosized nature of the g-C₃N₄ QDs, further in vivo safety and efficacy testing are required to ensure they pose no risk to human. The regulatory requirements for nanoparticle based medical products are highly stringent as the nanoparticles can travel in the body or enter the cells and their impacts must be evaluated.

Overall, I suggest that further in vitro and in vivo testing (as mentioned above) are required to make the manuscript publishable in Nature Communications.

RESPONSE TO REVIEWER COMMENTS

Reviewer #1 (Remarks to the Author):

The manuscript by Yang et al highlights a very interesting biomedical approach of a nano-photocatalyst- g-C₃N₄ QDs. Keratoconus is a challenging clinical condition and proposing g-C₃N₄ QDs for corneal crosslinking is a novel approach for the field.

Response: Thank you very much for your thoughtful comment and valuable suggestions. We are truly grateful for your interest in our work. This work represents the first report of using g-C₃N₄ QDs as a kind of oxygen self-producing and self-consuming photosensitizer effectively promotes corneal cross-linking effect, paving the way for a new type of photosensitizer for accelerated corneal crosslinking. For your convenience, your comments have been categorized into six sections and addressed individually. The detailed responses are presented as follows.

When a nanomaterial is tested for a biomedical application, it would be really important to assess the stability of the material. What is the stability profile of g-C₃N₄ QDs over time? How does biological environment/fluid change the stability of this material, considering its suggested use for cornea? Does the nanomaterial aggregate or precipitate over time before application?

Response: Thank you for your valuable comment. The obtained g-C₃N₄ QDs predominantly consist of nanoparticles ranging in diameter from 5 to 8 nm. Previous experiments conducted by our team have shown that an aqueous dispersion of these g-C₃N₄ QDs, prepared at a concentration of 2.1 mg/mL and stored in a 4°C, retains its transparency without any visible precipitation for approximately 3 months. This observation can serve as testament to the remarkable dispersibility and stability of the obtained g-C₃N₄ QDs. More importantly, the aqueous dispersion of the obtained g-C₃N₄ QDs can be converted into a powder form through lyophilization, enabling long-term storage. When redispersed in water, this powder readily forms a transparent aqueous dispersion suitable for a variety of applications, including A-CXL. The uniform redispersion of the lyophilized g-C₃N₄ QD powder in water, resulting in a transparent dispersion, further underscores its excellent dispersibility. The corresponding data are presented in Figure S1 as below. We have included the new findings in the "**Synthesis and Characterization of g-C₃N₄ QDs**" section of the **Results and Discussion** section, as shown below.

"After synthesis, a light yellow transparent g-C₃N₄ QDs aqueous dispersion can be obtained, which exhibits excellent dispersibility in water even been lyophilized into g-C₃N₄ QDs powder (as shown in Figure S1)."

Fig. S1 Photographs of the g-C₃N₄ QDs aqueous dispersion and re-dispersed aqueous dispersion.

For cornea, transparency is an essential parameter. On the other hand, full dispersibility of nanomaterial is a limiting factor. What is the effect of the g-C₃N₄ QDs on the corneal transparency?

Response: Thanks for your useful comment. Following your recommendation, the deepithelialized cornea that had undergone a 30-minute treatment with the g-C₃N₄ QDs aqueous dispersion was harvested and subjected to spectrophotometric analysis using a UV/Vis spectrophotometer. The findings revealed that the transmittance of the treated corneas closely mirrored that of the untreated control. Additionally, the photographs of the treated cornea were also collected and juxtaposed with the untreated cornea for comparison. As exemplified in Figure S11, the distinct visibility of the "E" logo is indicative of the superior transparency achieved in the treated cornea, thereby suggesting a negligible impact of the g-C₃N₄ QDs on corneal transparency.

Fig. S11 The comparison of the transparency of the deepithelialized cornea presoaked for 30 min using 2.5 mg mL⁻¹ g-C₃N₄ QDs aqueous dispersion with untreated cornea. (a) transmittance of the cornea with light wavelength controlled at 510 nm. (b) Photographs of the corneas with different conditions.

More emphasis can be made for material choice. Why did the authors select this nano photocatalyst out of others available in literature? Why did they select the quantum dot version instead of the nanosheet format? How is it compared with other nano based corneal crosslinking studies?

Response: One of the purposes of this article is to address the issue of hypoxia encountered during the corneal crosslinking process. Given the abundance of water molecules in corneal stroma layer, photocatalysts, which are used in the photocatalytic decomposition of water to generate oxygen, could offer a solution to the problem. Notably, g-C₃N₄ has garnered attention for its remarkable photocatalytic ability to decompose water into oxygen, besides its non-toxic, appropriate electron energy level structure, stable physical and chemical properties. However, the majority of industrially produced g-C₃N₄ particles are obtained through direct calcination, resulting in size on micrometers scale, which are unsuitable for biomedical applications.

Therefore, we developed a novel type of hydrophilic g-C₃N₄ QDs with size ranging from 5-8 nm. Their small size facilitates dispersion and metabolism within the stromal layer. In addition, these g-C₃N₄ QDs can also function as photosensitizers, capable of producing singlet oxygen under ultraviolet light irradiation, thus enhancing corneal cross-linking effect. Our experimental results demonstrated that this kind of oxygen self-producing and self-consuming photosensitizer effectively promotes corneal cross-linking effect, paving the way for a new type of photosensitizer for accelerated corneal crosslinking.

Up to now, among kinds of photosensitizers used for corneal cross-linking, riboflavin is still the focus of current research and clinical application, other photosensitizers such as Rose Bengal have also been reported. In this article, the performance of g-C₃N₄ QDs as a photosensitizer for corneal cross-linking was studied for the first time, which displays excellent accelerated corneal crosslinking effect, comparable to the cross-linking effect of classical deepithelial riboflavin experimental group, is worthy for further research and exploration.

In the results section, it might be beneficial to delve into the clinical relevance of your study's findings. Specifically, consider explaining how the observed biocompatibility of g-C₃N₄ QDs within the cornea is directly linked to potential applications in clinical practice, particularly in the context of corneal cross-linking (A-CXL).

Response: Thanks for your instrumental suggestion. Following your comment, the short- and long-term biocompatibility of g-C₃N₄ QDs in vivo was evaluated through a series of comprehensive tests, including slit lamp biomicroscopy for detailed ocular examination, specular microscope evaluation to assess endothelial cell health, corneal hematoxylin and eosin (H&E) staining for histological analysis, alizarin red S and trypan blue endothelial staining (ES) to visualize the endothelium, corneal TUNEL staining to detect apoptotic cells, and other comprehensive ocular examinations. Additionally, we conducted routine blood analysis, biochemical tests, and hematoxylin and eosin staining of various organs at designated intervals following A-CXL treatment, besides the hemolysis experiment of g-C₃N₄ QDs.

The results of our evaluation were presented in the revised manuscript and accompanying supporting information (Figures 7, 8, and Figures S15-S18) as below. These findings provided additional evidence for the excellent in vivo biological safety of the prepared g-C₃N₄ QDs. The new findings have been incorporated into the "**The eye biocompatibility of g-C₃N₄ QDs**", and "**The in vivo safety of g-C₃N₄ QDs**" sections of the **Results and Discussion** section, as shown below.

"The eye biocompatibility of g-C₃N₄ QDs

The biocompatibility of g-C₃N₄ QDs in the cornea during various CXL treatments was also studied. Prior to CXL treatment, an ocular surface irritation evaluation was conducted on rabbits. Utilizing slit lamp

microscopy, the integrity of the corneal epithelium was carefully observed. Photographic documentation was collected at 6 h and 24 h post-administration. As illustrated in Fig. 7a, when compared to the PBS control group, no staining of the corneal epithelium or epithelial defect were detected following fluorescein sodium staining. Meanwhile, there were no apparent symptoms of anterior ganglia irritation. Specifically, the cornea remained transparency without turbidity, the conjunctiva showed no signs of congestion or edema, there was no iris involvement, and no abnormal secretions were present within the 24-hour observation period. These findings indicate that the obtained $g\text{-C}_3\text{N}_4$ QDs dispersion exhibits weak ocular surface irritation.

After the application of A-CXL treatment, corneal recovery was initially monitored using a slit-lamp microscope, with the RF 9 mW cm^{-2} group serving as a control. As shown in Fig. S15, during the initial days following the treatment, both groups exhibited notable signs of anterior ganglion irritation, corneal opacity, conjunctival hyperemia and edema, iris involvement, and abnormal secretions. Notably, these symptoms appeared to be more pronounced in the RF 9 mW cm^{-2} group. These manifestations were primarily attributed to the removal of the corneal epithelium and gradually subsided over time. By the 7th day post-treatment, a significant recovery of ocular tissues was observed for $g\text{-C}_3\text{N}_4$ QDs group. By the 14th day, the eye had essentially reverted to its pre-surgical state. As time progressed, minimal to no discernible differences can be observed in the cornea upon slit-lamp examination.

Meanwhile, the corneal endothelium of rabbits subjected to various treatments was evaluated using a specular microscope (EM-4000). Data on the endothelial cell count from different groups post-CXL treatment were gathered and analyzed. The results, as presented in Fig.7b, revealed a substantial decrease in the number of endothelial cells in the RF 3 mW cm^{-2} group following CXL (with significant difference of "*****"). In contrast, the A-CXL groups, particularly the $g\text{-C}_3\text{N}_4$ QDs group, demonstrated only minor reductions in endothelial cell count (with significant difference of "*"). These results suggest that A-CXL treatment, especially when utilizing $g\text{-C}_3\text{N}_4$ QDs as a photosensitizer, offers greater safety in terms of endothelial cell preservation (The observed overall decline in data across every experimental group might be attributed to the change of the rabbits' growth environment, the error margin of specular microscope, and potentially other variables). This conclusion is further corroborated by the images shown in Fig. S16, which exhibit well-maintained endothelial cell morphology and a smaller reduction in cell count in the treated eye compared to the untreated eye.

Moreover, the biocompatibility of $g\text{-C}_3\text{N}_4$ QDs on the cornea in vivo without and with A-CXL were also compared through histopathologic observations. Hematoxylin-Eosin staining of the cornea, Alizarin Red S and Trypan Blue staining of the endothelium as well as TUNEL staining were also carried out, and the results are shown in Fig. 7c-7e. The Hematoxylin-Eosin staining results (Fig. 7c) showed no obvious differences between the PBS group and the other irradiated and non-irradiated treatment groups, all cells in the stromal layer evenly distributed and no obvious cell damage was observed, indicating excellent biocompatibility of $g\text{-C}_3\text{N}_4$ QDs, even under the irradiation of 9 mW cm^{-2} UVA light for 10 min. Endothelium staining (ES) using Alizarin Red S and Trypan Blue were also carried out for the different groups without A-CXL and with A-CXL, and the results are shown in Fig. 7d. Like the PBS and RF 9 mW cm^{-2} groups, no blue cells could be observed in the $g\text{-C}_3\text{N}_4$ QDs group, indicating that all cells remain healthy, even after undergoing A-CXL. The intact endothelial structure further indicates the healthy state of the cells in the $g\text{-C}_3\text{N}_4$ QDs group without and with A-CXL. This conclusion was further confirmed by the quantitative measurements from the ES photos (insert in Fig. 7d). TUNEL staining of cornea was also carried out to assay the apoptosis effects of the synthesized $g\text{-C}_3\text{N}_4$ QDs, and the results are shown in Fig.7e with the standard positive and negative group displayed in Fig. S17. No green color can be observed in the images of the $g\text{-C}_3\text{N}_4$ QDs group without A-CXL, indicating that the synthesized $g\text{-C}_3\text{N}_4$ QDs did not induce apoptosis without A-CXL, confirming the excellent biocompatibility of the synthesized $g\text{-C}_3\text{N}_4$ QDs. Similar with the samples without A-CXL, seldom green spots can be observed in the images of the different groups with A-CXL, indicating that the 10 min's irradiation of 9 mW cm^{-2} 365 nm UVA light on the cornea has little harm either the epithelium, the stroma, or the endothelium in vivo, providing significance information for the potential clinical applications.

The in vivo safety of $g\text{-C}_3\text{N}_4$ QDs

Simultaneously, the in vivo safety of the proposed $g\text{-C}_3\text{N}_4$ QDs was also assessed. Initially, an assessment of the hemolysis potential of the $g\text{-C}_3\text{N}_4$ QDs dispersion was conducted, and the outcomes were displayed in Fig.8a and Fig. S18. The observations revealed that even at a concentration of 800 $\mu\text{g mL}^{-1}$, the hemolysis rate of red blood cells remained low, approximately 0.6%. This finding suggests that the obtained $g\text{-C}_3\text{N}_4$ QDs had minimal influence on immune regulation or heme response, thereby exhibited good blood biocompatibility.

Besides, blood samples were collected from rabbits at 1, 15, and 30 days post A-CXL treatment using $g\text{-C}_3\text{N}_4$ QDs dispersion. Routine blood analysis and biochemistry tests were conducted on these samples, with untreated rabbits serving as controls. The results of these tests are presented in Fig. 8b-8d. In comparison to the control group, three indicators including ALT, ALB, PLT, exhibited abnormalities on both day 1 and day 15 following A-CXL treatment. Nevertheless, by the thirtieth day, these indicators had reverted to normal levels. It is noteworthy that all other blood parameters showed no statistically significant differences compared to the control group. Taken together, these findings collectively indicate that there was minimal to no significant inflammation or infection in rabbits treated with the synthesized $g\text{-C}_3\text{N}_4$ QDs as photosensitizers. The results strongly support the safety and potential clinical applicability of $g\text{-C}_3\text{N}_4$ QDs as promising A-CXL agents.

At the same time, multiple tissues, including the heart, liver, spleen, lung, and kidney, were collected at 30 days post A-CXL treatment using PBS, RF solution and $g\text{-C}_3\text{N}_4$ QDs dispersion. H&E staining were

carried out, with untreated rabbits serving as controls. As depicted in Fig. 8e, the results demonstrated that there were no observable pathological changes in the heart, liver, spleen, lung, or kidney over the course of treatment. Specifically, hepatocytes appeared normal in the liver samples, no pulmonary fibrosis was detected in the lung samples, the glomerulus structure was clearly visible in the kidney sections, and there was no evidence of necrosis in any of the histological samples. These findings strongly suggest that g-C₃N₄ QDs have minimal side effects and possess great clinical applicability as A-CXL agents."

Fig. 7 The ocular biocompatibility of g-C₃N₄ QDs without or with A-CXL. (a) Slit lamp images of the eyes treated with g-C₃N₄ QDs. (b) Corneal endothelial cell density on the 0, 1, 15, and 30 days (mean ± SD, n≥3). (c) Hematoxylin-Eosin staining (H&E) of the cornea. (d) Endothelial staining (ES) and the statistical analysis of ES photos (insert in Fig. 7b). (e) TUNEL test of cornea after being soaked with PBS, RF 9 mW cm⁻², and g-C₃N₄ QDs for 30 min. Data are presented as (mean ± SD, n≥3). Statistical analysis was performed using one-way ANOVA with Tukey multiple comparisons test, ns: no significance.

Fig. 8 The systemic biocompatibility of g-C₃N₄ QDs. (a) Hemolysis experimental analysis of g-C₃N₄ QDs. (b-d) Routine blood analysis and biochemistry tests of the rabbits in g-C₃N₄ QDs group on days 0, 1, 15, and 30 days (WBC: White Blood cell count; Lymph#: Lymphocyte count; Mon#: monocyte count; Gran#: Granulocyte count; Lymph: Lymphocyte percentage; Mon: monocyte percentage; Gran: Granulocyte percentage; RBC: Red Blood cell count; HCT: Hematocrit; MCV: Mean corpuscular volume; MCH: Mean corpuscular hemoglobin; RDW: Red cell distribution width; MPV: Mean platelet volume; PDW: Platelet distribution width; PCT: Plateletcrit; ALT: Alanine transaminase; ALB: Albumin/Albumin blood test; REA: Urea/urea test; AST: Aspartate aminotransferase; ALP: Alkaline phosphatase; CREA: Creatinine/creatinine test; TBIL: Total bilirubin; γ -GT: Gamma-glutamyl transferase; DBIL: Direct bilirubin; TBA: Total bile acid/Total bile acid test; UA: Uric acid/uric acid test; HGB: Hemoglobin; MCHC: Mean corpuscular hemoglobin concentration; PLT: Platelets count). (e) Hematoxylin-Eosin staining (H&E) of the different organs of the rabbits in g-C₃N₄ QDs and RF 9 mW cm⁻² groups on days 30. Data are presented as mean \pm SD, n \geq 3. ns: no significance, *p < 0.05, **p < 0.01, ***p < 0.001, ****p < 0.0001.

Fig. S15 The postoperative reaction and recovery condition of the rabbits' eye after A-CXL with different photosensitizer. DI: The photographs were taken using diffuse illumination (DI) for observation of the anterior segment overview. NSB: The photographs were taken using narrow slit beam (NSB) with background illumination for observation of corneal thickness.

Fig. S16 The photos of the in vivo corneal endothelium in g-C₃N₄ QDs groups after A-CXL for different observation time.

Fig. S17 The confocal images of negative group (normal tissue) and positive groups (DNase-treated) in TUNEL testing.

Fig. S18 The hemolytic reaction of red blood cells incubated with water, PBS, or g-C₃N₄ QDs for 2 h.

How will this application contribute to the future of the cornea department, and does it have the potential to enable office-based corneal cross-linking (CXL)? How is g-C₃N₄ QD based crosslinking compared to clinically applied strategies?

Response: Thanks for your helpful comment. Currently, riboflavin remains the primary photosensitizer employed in clinical corneal crosslinking, yet numerous challenges persist during the crosslinking procedure. One challenge is the insufficient oxygen content encountered during the process, which diminishes the effectiveness of transepithelial corneal crosslinking and accelerated corneal crosslinking with riboflavin (insufficient riboflavin is another reason). Consequently, patients frequently experience disease progression post-surgery. The introduction of g-C₃N₄ QDs as a photosensitizer in corneal crosslinking is expected to solve the hypoxia problem during the crosslinking process. This photosensitizer has demonstrated remarkable crosslinking effects in hypoxic conditions. Our animal studies have revealed that g-C₃N₄ QDs exhibit an excellent accelerated corneal crosslinking effect, comparable to the crosslinking effect of classical deepithelial riboflavin experimental group. It is worth for further exploration and research, which may open up a new direction and provide new ideas for the development of corneal cross-linking photosensitizer.

It appears that there is a typographical error in Figure 6's footnote. The correct term should be "Tukey" multiple comparisons test, not "Turkey." Here's the corrected footnote: Figure 6 footnote: "Statistical analysis was performed using one-way ANOVA with Tukey multiple comparisons test"

Response: Thank you for pointing this out. We apologize for the mistake we made, the corresponding part has been corrected in the revised manuscript, which has been highlighted in yellow.

Reviewer #2 (Remarks to the Author):

This work is novel and may have a significant positive impact on the field of cornea and collagen crosslinking. It is previously reported that additional supply of ex-situ oxygen to the cornea (e.g., outside the cornea via goggles, facemask) increases the crosslinking yield. What is noteworthy about this work is that the oxygen supply is done in-situ (e.g. inside the cornea) enhancing availability of oxygen at the crosslinking site. Despite the fact that the concept is novel, the experimental work do not fully support the conclusions and additional evidence is needed, for instance:

Response: We very much appreciate your valuable comment. Following your helpful suggestion, a large number of experiments were further performed. The subsequent findings have been meticulously included in the revised manuscript and accompanying supporting information, which have been highlighted in yellow for your convenience. Once again, we sincerely thank you very much for your time and efforts in helping us improve our work. The point-by-point response is shown below.

1. The HCEC cells shown in Fig 3 (a) (in vitro cell viability study) seem round and not elongated suggesting poor cell biocompatibility. Also, by increasing the g-C₃N₄ QDs concentration, the number of live cells seem to decrease while not obvious in Fig.3 (e). This may be due to poor quality of the images.

Response: Thanks for your meaningful comment. Acting upon your suggestion, we have reconducted the Calcein-AM/PI double staining and Flow-cytometric analysis of HCEC cells, with the revised outcomes reflected in Figure 3d,3e, which are also displayed as following. Notably, consistent observations were made, underscoring the remarkable cell biocompatibility of g-C₃N₄ QDs, even at elevated concentrations of g-C₃N₄ QDs to 400 mg/mL. It is worth mentioning that the subtle rounding of cell morphology captured in the Calcein-AM/PI double staining images might be due to an extended period of cell retention following the double staining procedure prior to analysis, leading to a compromised cell condition.

Fig. 3. In vitro biocompatible evaluation of g-C₃N₄ QDs. (d) Calcein-AM/PI double staining of HCEC

cells incubated with g-C₃N₄ QDs at different concentrations. (e, f) Flow-cytometric analysis of HCEC cells incubated with g-C₃N₄ QDs at different concentrations.

2. the authors have used their g-C₃N₄ QDs in vivo in rabbits, however the degradation and mechanical tests are done on ex-vivo samples and no in vivo testing such as biomechanics (e.g., Corvis® ST) OCT for thickness measurement, or in vivo toxicity tests are not performed. Given the nanosized nature of the g-C₃N₄ QDs, further in vivo safety and efficacy testing are required to ensure they pose no risk to human. The regulatory requirements for nanoparticle based medical products are highly stringent as the nanoparticles can travel in the body or enter the cells and their impacts must be evaluated.

Response: Thanks for your ponderable comment. Following your suggestion, we conducted a comprehensive array of in vivo tests on rabbits after corneal cross-linking with g-C₃N₄ QDs serving as the photosensitizer. Our investigations included in vivo biomechanics measurement, the short- and long-term biocompatibility of g-C₃N₄ QDs in vivo. Specifically, we employed Corvis® ST measurements to assess biomechanics, OCT measurements to determine cornea thickness, slit lamp biomicroscopy for detailed ocular examination, specular microscope evaluation to assess endothelial cell health, corneal hematoxylin and eosin (H&E) staining for histological analysis, alizarin red S and trypan blue endothelial staining (ES) to visualize the endothelium, corneal TUNEL staining to detect apoptotic cells, and other comprehensive ocular examinations. Additionally, we conducted routine blood analysis, biochemical tests, and hematoxylin and eosin staining of various organs at designated intervals following A-CXL treatment, besides the hemolysis experiment of g-C₃N₄ QDs.

The results obtained from these tests have been meticulously incorporated into Figure 6-8 and Figure S14-S18 as below. The new findings have been included in the "**A-CXL Evaluations of g-C₃N₄ QDs**", "**The eye biocompatibility of g-C₃N₄ QDs**", and "**The in vivo safety of g-C₃N₄ QDs**" sections of the **Results and Discussion** section, as shown below.

"Concurrently, an evaluation of the in vivo biomechanical properties of the cornea was conducted following CXL treatment with various photosensitizers at different time points (1, 3, 5, 7, 14, 21 days) using Corvis ST. Key corneal biomechanical parameters, including the first applanation length (A1L), the second applanation velocity (A2V), the radius at highest concavity (HC-R), and deformation amplitude at highest concavity (HC-DA), were selected and compared with those of normal corneas (PBS group). Among the parameters measured by Corvis ST, the flattening speed serves as a proxy for the overall stiffness and resistance of the cornea. As corneal stiffness increases, so does its resistance, resulting in a reduced compression speed in response to airflow. The maximum deformation amplitude of the cornea, on the other hand, reflects its softness. The harder the cornea, the smaller deformation amplitude. As shown in Fig. 6b-6e, corneal edema, resulting from the removal of corneal epithelium, can influence the data collected within the initial few days, leading to some discrepancies when compared to other groups. Nonetheless, statistically significant differences can be observed in the A1L values of the corneas in the g-C₃N₄ QDs group when compared to both the PBS and RF 9 mW cm⁻² groups under identical conditions. These values are comparable to those obtained using the conventional RF CXL protocol (RF 3 mW cm⁻²) with no significant difference, indicating a positive A-CXL effect when using g-C₃N₄ QDs as a photosensitizer (Fig.6b). Furthermore, the observed slower A2V values (Fig.6c), the increase in HC-R (Fig.6d), and the smaller HC-DA values (Fig.6e) observed in the corneas of the g-C₃N₄ QDs group suggest an improvement in corneal biomechanical properties following A-CXL using g-C₃N₄ QDs as a photosensitizer. Collectively, these findings underscore the potential of g-C₃N₄QDs as effective photosensitizers in A-CXL treatments aimed at enhancing corneal biomechanical properties.

Additionally, measurements of central corneal thickness (CCT) were also collected from the corneas in various groups at different time points (21, 30 days) following CXL treatment. These measurements are presented in Fig.6f, 6g and Fig. S14. As time progressed following CXL, corneal edema gradually abated. Notably, The CCT values for the corneas in both the g-C₃N₄ QDs and RF 3 mW cm⁻² groups approximated those recorded prior to A-CXL when the observation period was extended to 21 days. With a further prolongation of the observation period, minimal reductions in CCT values were observed (Fig. 6g and S14), indicating a faster recovery and healing effect of these two protocols. Nevertheless, it is worth mentioning that corneal edema persisted in a significant manner in the corneas of both the RF 9 mW cm⁻² and RF@g-C₃N₄ QDs groups even after 21 days. When the observation period was extended to 30 days, a distinct difference emerged between these two groups, unlike the RF 9 mW cm⁻² group, where the CCT values remained elevated compared to pre-CXL levels, the RF@g-C₃N₄ QDs group exhibited CCT values comparable to those recorded before CXL. Thus, the existence of g-C₃N₄ QDs may facilitate the recovery and healing of cornea.

The eye biocompatibility of g-C₃N₄ QDs

The biocompatibility of g-C₃N₄ QDs in the cornea during various CXL treatments was also studied. Prior to CXL treatment, an ocular surface irritation evaluation was conducted on rabbits. Utilizing slit lamp microscopy, the integrity of the corneal epithelium was carefully observed. Photographic documentation was collected at 6 h and 24 h post-administration. As illustrated in Fig. 7a, when compared to the PBS control group, no staining of the corneal epithelium or epithelial defect were detected following fluorescein sodium staining. Meanwhile, there were no apparent symptoms of anterior ganglia irritation. Specifically, the cornea remained transparency without turbidity, the conjunctiva showed no signs of congestion or edema, there

was no iris involvement, and no abnormal secretions were present within the 24-hour observation period. These findings indicate that the obtained $g-C_3N_4$ QDs dispersion exhibits weak ocular surface irritation.

After the application of A-CXL treatment, corneal recovery was initially monitored using a slit-lamp microscope, with the RF 9 mW cm^{-2} group serving as a control. As shown in Fig. S15, during the initial days following the treatment, both groups exhibited notable signs of anterior ganglion irritation, corneal opacity, conjunctival hyperemia and edema, iris involvement, and abnormal secretions. Notably, these symptoms appeared to be more pronounced in the RF 9 mW cm^{-2} group. These manifestations were primarily attributed to the removal of the corneal epithelium and gradually subsided over time. By the 7th day post-treatment, a significant recovery of ocular tissues was observed for $g-C_3N_4$ QDs group. By the 14th day, the eye had essentially reverted to its pre-surgical state. As time progressed, minimal to no discernible differences can be observed in the cornea upon slit-lamp examination.

Meanwhile, the corneal endothelium of rabbits subjected to various treatments was evaluated using a specular microscope (EM-4000). Data on the endothelial cell count from different groups post-CXL treatment were gathered and analyzed. The results, as presented in Fig. 7b, revealed a substantial decrease in the number of endothelial cells in the RF 3 mW cm^{-2} group following CXL (with significant difference of "****"). In contrast, the A-CXL groups, particularly the $g-C_3N_4$ QDs group, demonstrated only minor reductions in endothelial cell count (with significant difference of "**"). These results suggest that A-CXL treatment, especially when utilizing $g-C_3N_4$ QDs as a photosensitizer, offers greater safety in terms of endothelial cell preservation (The observed overall decline in data across every experimental group might be attributed to the change of the rabbits' growth environment, the error margin of specular microscope, and potentially other variables). This conclusion is further corroborated by the images shown in Fig. S16, which exhibit well-maintained endothelial cell morphology and a smaller reduction in cell count in the treated eye compared to the untreated eye.

Moreover, the biocompatibility of $g-C_3N_4$ QDs on the cornea in vivo without and with A-CXL were also compared through histopathologic observations. Hematoxylin-Eosin staining of the cornea, Alizarin Red S and Trypan Blue staining of the endothelium as well as TUNEL staining were also carried out, and the results are shown in Fig. 7c-7e. The Hematoxylin-Eosin staining results (Fig. 7c) showed no obvious differences between the PBS group and the other irradiated and non-irradiated treatment groups, all cells in the stromal layer evenly distributed and no obvious cell damage was observed, indicating excellent biocompatibility of $g-C_3N_4$ QDs, even under the irradiation of 9 mW cm^{-2} UVA light for 10 min. Endothelium staining (ES) using Alizarin Red S and Trypan Blue were also carried out for the different groups without A-CXL and with A-CXL, and the results are shown in Fig. 7d. Like the PBS and RF 9 mW cm^{-2} groups, no blue cells could be observed in the $g-C_3N_4$ QDs group, indicating that all cells remain healthy, even after undergoing A-CXL. The intact endothelial structure further indicates the healthy state of the cells in the $g-C_3N_4$ QDs group without and with A-CXL. This conclusion was further confirmed by the quantitative measurements from the ES photos (insert in Fig. 7d). TUNEL staining of cornea was also carried out to assay the apoptosis effects of the synthesized $g-C_3N_4$ QDs, and the results are shown in Fig. 7e with the standard positive and negative group displayed in Fig. S17. No green color can be observed in the images of the $g-C_3N_4$ QDs group without A-CXL, indicating that the synthesized $g-C_3N_4$ QDs did not induce apoptosis without A-CXL, confirming the excellent biocompatibility of the synthesized $g-C_3N_4$ QDs. Similar with the samples without A-CXL, seldom green spots can be observed in the images of the different groups with A-CXL, indicating that the 10 min's irradiation of 9 mW cm^{-2} 365 nm UVA light on the cornea has little harm either the epithelium, the stroma, or the endothelium in vivo, providing significance information for the potential clinical applications.

The in vivo safety of $g-C_3N_4$ QDs

Simultaneously, the in vivo safety of the proposed $g-C_3N_4$ QDs was also assessed. Initially, an assessment of the hemolysis potential of the $g-C_3N_4$ QDs dispersion was conducted, and the outcomes were displayed in Fig. 8a and Fig. S18. The observations revealed that even at a concentration of 800 $\mu g mL^{-1}$, the hemolysis rate of red blood cells remained low, approximately 0.6%. This finding suggests that the obtained $g-C_3N_4$ QDs had minimal influence on immune regulation or heme response, thereby exhibited good blood biocompatibility.

Besides, blood samples were collected from rabbits at 1, 15, and 30 days post A-CXL treatment using $g-C_3N_4$ QDs dispersion. Routine blood analysis and biochemistry tests were conducted on these samples, with untreated rabbits serving as controls. The results of these tests are presented in Fig. 8b-8d. In comparison to the control group, three indicators including ALT, ALB, PLT, exhibited abnormalities on both day 1 and day 15 following A-CXL treatment. Nevertheless, by the thirtieth day, these indicators had reverted to normal levels. It is noteworthy that all other blood parameters showed no statistically significant differences compared to the control group. Taken together, these findings collectively indicate that there was minimal to no significant inflammation or infection in rabbits treated with the synthesized $g-C_3N_4$ QDs as photosensitizers. The results strongly support the safety and potential clinical applicability of $g-C_3N_4$ QDs as promising A-CXL agents.

At the same time, multiple tissues, including the heart, liver, spleen, lung, and kidney, were collected at 30 days post A-CXL treatment using PBS, RF solution and $g-C_3N_4$ QDs dispersion. H&E staining were carried out, with untreated rabbits serving as controls. As depicted in Fig. 8e, the results demonstrated that there were no observable pathological changes in the heart, liver, spleen, lung, or kidney over the course of treatment. Specifically, hepatocytes appeared normal in the liver samples, no pulmonary fibrosis was detected in the lung samples, the glomerulus structure was clearly visible in the kidney sections, and there

was no evidence of necrosis in any of the histological samples. These findings strongly suggest that $g\text{-C}_3\text{N}_4$ QDs have minimal side effects and possess great clinical applicability as A-CXL agents."

Fig. 6 The corneal structure variation after A-CXL treatment. (a) Schematic of a simple operation flow chart of A-CXL evaluation using Corvis ST and Optovue RTVue OCT. (b) Comparison of A1L (mean \pm SD, $n \geq 3$). (c) Comparison of A2V (mean \pm SD, $n \geq 3$). (d) Comparison of HC-R (mean \pm SD, $n \geq 3$). (e) Comparison of HC-DA (mean \pm SD, $n \geq 3$). (f) Photos of OCT images, (g) Comparison of the CCT (mean \pm SD, $n \geq 3$). ns: no significance. * $p < 0.05$, ** $p < 0.01$, *** $p < 0.001$, **** $p < 0.0001$.

Fig. 7 The ocular biocompatibility of g-C₃N₄ QDs without or with A-CXL. (a) Slit lamp images of the eyes treated with g-C₃N₄ QDs. (b) Corneal endothelial cell density on the 0, 1, 15, and 30 days (mean ± SD, n≥3). (c) Hematoxylin-Eosin staining (H&E) of the cornea. (d) Endothelial staining (ES) and the statistical analysis of ES photos (insert in Fig. 7b). (e) TUNEL test of cornea after being soaked with PBS, RF 9 mW cm⁻², and g-C₃N₄ QDs for 30 min. Data are presented as (mean ± SD, n≥3). Statistical analysis was performed using one-way ANOVA with Tukey multiple comparisons test, ns: no significance.

Fig. 8 The systemic biocompatibility of g-C₃N₄ QDs. (a) Hemolysis experimental analysis of g-C₃N₄ QDs. (b-d) Routine blood analysis and biochemistry tests of the rabbits in g-C₃N₄ QDs group on days 0, 1, 15, and 30 days (WBC: White Blood cell count; Lymph#: Lymphocyte count; Mon#: monocyte count; Gran#: Granulocyte count; Lymph: Lymphocyte percentage; Mon: monocyte percentage; Gran: Granulocyte percentage; RBC: Red Blood cell count; HCT: Hematocrit; MCV: Mean corpuscular volume; MCH: Mean corpuscular hemoglobin; RDW: Red cell distribution width; MPV: Mean platelet volume; PDW: Platelet distribution width; PCT: Plateletcrit; ALT: Alanine transaminase; ALB: Albumin/Albumin blood test; REA: Urea/urea test; AST: Aspartate aminotransferase; ALP: Alkaline phosphatase; CREA: Creatinine/creatinine test; TBIL: Total bilirubin; γ -GT: Gamma-glutamyl transferase; DBIL: Direct bilirubin; TBA: Total bile acid/Total bile acid test; UA: Uric acid/uric acid test; HGB: Hemoglobin; MCHC: Mean corpuscular hemoglobin concentration; PLT: Platelets count). (e) Hematoxylin-Eosin staining (H&E) of the different organs of the rabbits in g-C₃N₄ QDs and RF 9 mW cm⁻² groups on days 30. Data are presented as mean \pm SD, $n \geq 3$. ns: no significance, * $p < 0.05$, ** $p < 0.01$, *** $p < 0.001$, **** $p < 0.0001$.

Fig. S14 OCT images of cornea in $g\text{-C}_3\text{N}_4$ QDs group at different time intervals after A-CXL.

Fig. S15 The postoperative reaction and recovery condition of the rabbits' eye after A-CXL with different photosensitizer. DI: The photographs were taken using diffuse illumination (DI) for observation of the anterior segment overview. NSB: The photographs were taken using narrow slit beam (NSB) with background illumination for observation of corneal thickness.

Fig. S16 The photos of the in vivo corneal endothelium in g-C₃N₄ QDs groups after A-CXL for different observation time.

Fig. S17 The confocal images of negative group (normal tissue) and positive groups (DNase-treated) in TUNEL testing.

Fig. S18 The hemolytic reaction of red blood cells incubated with water, PBS, or g-C₃N₄ QDs for 2 h.

Overall, I suggest that further in vitro and in vivo testing (as mentioned above) are required to make the manuscript publishable in Nature Communications.

Response: Following your valuable suggestion, a comprehensive suite of experimental work was carried out (as mentioned above). The obtained results have been meticulously incorporated into the revised manuscript and corresponding supporting information, which have been highlighted in yellow for your convenience.

Reviewers' Comments:

Reviewer #1:

Remarks to the Author:

Authors performed a comprehensive review of the manuscript. With the additional in vitro and in vivo experiments, reviewer questions are mostly answered and the quality of the manuscript has been substantially improved. One remaining point is related to my first question: "..What is the stability profile of g-C₃N₄ QDs over time? How does biological environment/fluid change the stability of this material, considering its suggested use for cornea? Does the nanomaterial aggregate or precipitate over time before application?". Although authors provided a photograph of the g-C₃N₄ QDs aqueous dispersion and re-dispersed aqueous dispersion, it may not fully answer the raised question. It is important to show the change of size over time in PBS/water and serum, as done with any nanoparticles designed for biomedical applications. After this change, the manuscript will be suitable for publication.

Reviewer #2:

Remarks to the Author:

The authors have addressed the shortcoming of the first article submission in the revised version. As such, I suggest accepting the article for publication.

RESPONSES TO REVIEWERS' COMMENTS

Reviewer #1 (Remarks to the Author):

Authors performed a comprehensive review of the manuscript. With the additional *in vitro* and *in vivo* experiments, reviewer questions are mostly answered and the quality of the manuscript has been substantially improved.

One remaining point is related to my first question: "What is the stability profile of g-C₃N₄ QDs over time? How does biological environment/fluid change the stability of this material, considering its suggested use for cornea? Does the nanomaterial aggregate or precipitate over time before application?". Although authors provided a photograph of the g-C₃N₄ QDs aqueous dispersion and re-dispersed aqueous dispersion, it may not fully answer the raised question. It is important to show the change of size over time in PBS/water and serum, as done with any nanoparticles designed for biomedical applications. After this change, the manuscript will be suitable for publication.

Response: Thank you very much for your thoughtful comment and valuable suggestions. We are truly grateful for your interest in our work. Following your suggestion, the stability of the synthesized g-C₃N₄ QDs dispersed in various solutions, such as deionized water (H₂O), PBS, 10% fetal bovine serum (FBS) aqueous solution (10% FBS), and 100% FBS, was studied. Since the samples can be lyophilized into g-C₃N₄ QDs powder and preserved for extended periods, only the state changes of the g-C₃N₄ QDs dispersed in these various solutions at 4 °C within 10-days timeframe were provided in this study. Photographs were captured from both the front view (FV) and the top view (TV) of the dispersions, with the side of the dispersions irradiated by laser light to exhibit the Tyndall effect. As depicted in Supplementary Figure 11, the g-C₃N₄ QDs dispersed in the four solvents retained their transparency and clarity over time, indicating that the synthesized g-C₃N₄ QDs maintain good dispersibility and stability in different solutions. Particle size test results of the g-C₃N₄ QDs dispersed in these solutions further confirmed their good stability and dispersibility. However, the hydrated particle size of the samples varies in different dispersions, possibly due to different solvent effect, as illustrated in Supplementary Figure 12.

Supplementary Figure 11. The stability evaluation of the synthesized g-C₃N₄ QDs dispersed in deionized water (H₂O), PBS, 10% fetal bovine serum (FBS) aqueous solution (10% FBS), and 100% FBS.

Supplementary Figure 12. The particle size distribution statistics patterns of the g-C₃N₄ QDs dispersed in deionized water (H₂O), PBS, 10% fetal bovine serum (FBS) aqueous solution (10% FBS), and 100% FBS solution over a period of 10 days.

Reviewer #2 (Remarks to the Author):

The authors have addressed the shortcoming of the first article submission in the revised version. As such, I suggest accepting the article for publication.

Response: Thank you for your valuable comment. We are truly grateful for your engagement with our work and are pleased that you are satisfied with our revised version.

Reviewers' Comments:

Reviewer #1:

Remarks to the Author:

Authors have performed detailed stability assays. The manuscript is now suitable for publication.
As a final remark, please add the x axis label on the Supplementary Figure 12.

RESPONSE TO REVIEWER COMMENTS

Reviewer #1 (Remarks to the Author):

Authors have performed detailed stability assays. The manuscript is now suitable for publication. As a final remark, please add the x axis label on the Supplementary Figure 12.

Reply: Thank you for your valuable comment. We sincerely appreciate your engagement with our work and are pleased that you are satisfied with our revised version. Following your suggestion, the x axis label has been included in Supplementary Figure 12 of the revised manuscript, presented as follows:

Supplementary Figure 12. The particle size distribution statistics patterns of the g-C₃N₄ QDs dispersed in deionized water (H₂O), PBS, 10% fetal bovine serum (FBS) aqueous solution (10% FBS), and 100% FBS solution over a period of 10 days.